# Measuring Compositional Generalization: A Comprehensive Method on Realistic Data

**Daniel Keysers, Nathanael Schärli, Nathan Scales, Hylke Buisman, Daniel Furrer, Sergii Kashubin, Nikola Momchev, Danila Sinopalnikov, Lukasz Stafiniak, Tibor Tihon, Dmitry Tsarkov, Xiao Wang, Marc van Zee & Olivier Bousquet**

Google Research, Brain Team

```
{keysers,schaerli,nkscales,hylke,danielfurrer,sergik,nikola,sinopalnikov,
lukstafi,ttihon,tsar,wangxiao,marcvanzee,obousquet}@google.com
```

## Abstract

State-of-the-art machine learning methods exhibit limited *compositional generalization*. At the same time, there is a lack of *realistic* benchmarks that *comprehensively* measure this ability, which makes it challenging to find and evaluate improvements. We introduce a novel method to systematically construct such benchmarks by maximizing compound divergence while guaranteeing a small atom divergence between train and test sets, and we quantitatively compare this method to other approaches for creating compositional generalization benchmarks. We present a large and realistic natural language question answering dataset that is constructed according to this method, and we use it to analyze the compositional generalization ability of three machine learning architectures. We find that they fail to generalize compositionally and that there is a surprisingly strong negative correlation between compound divergence and accuracy. We also demonstrate how our method can be used to create new compositionality benchmarks on top of the existing SCAN dataset, which confirms these findings.

## 1 Introduction

Human intelligence exhibits *systematic compositionality* (Fodor & Pylyshyn, 1988), the capacity to understand and produce a potentially infinite number of novel combinations of known components, i.e., to make "infinite use of finite means" (Chomsky, 1965). In the context of learning from a set of training examples, we can observe compositionality as *compositional generalization*, which we take to mean the ability to systematically generalize to composed test examples of a certain distribution after being exposed to the necessary components during training on a different distribution.

Humans demonstrate this ability in many different domains, such as natural language understanding (NLU) and visual scene understanding. For example, we can learn the meaning of a new word and then apply it to other language contexts. As Lake & Baroni (2018) put it: "Once a person learns the meaning of a new verb 'dax', he or she can immediately understand the meaning of 'dax twice' and 'sing and dax'." Similarly, we can learn a new object shape and then understand its compositions with previously learned colors or materials (Johnson et al., 2017; Higgins et al., 2018).

In contrast, state-of-the-art machine learning (ML) methods often fail to capture the compositional structure that is underlying the problem domain and thus fail to generalize compositionally (Lake & Baroni, 2018; Bastings et al., 2018; Loula et al., 2018; Russin et al., 2019; Johnson et al., 2017). We believe that part of the reason for this shortcoming is a lack of realistic benchmarks that comprehensively measure this aspect of learning in realistic scenarios.

As others have proposed, compositional generalization can be assessed using a train-test split based on observable properties of the examples that *intuitively* correlate with their underlying compositional structure. Finegan-Dollak et al. (2018), for example, propose to test on different output patterns than are in the train set, while Lake & Baroni (2018) propose, among others, to split examples by output length or to test on examples containing primitives that are rarely shown during training. In this paper, we formalize and generalize this intuition and make these contributions:

- We introduce *distribution-based compositionality assessment (DBCA)*, which is a novel method to quantitatively assess the adequacy of a particular dataset split for measuring compositional generalization and to construct splits that are ideally suited for this purpose (Section 2).

- We present the Compositional Freebase Questions (CFQ)[1], a simple yet realistic and large NLU dataset that is specifically designed to measure compositional generalization using the DBCA method, and we describe how to construct such a dataset (Section 3).

- We use the DBCA method to construct a series of experiments for measuring compositionality on CFQ and SCAN (Lake & Baroni, 2018) and to quantitatively compare these experiments to other compositionality experiments (Section 4).

- We analyze the performance of three baseline ML architectures on these experiments and show that these architectures fail to generalize compositionally, and perhaps more surprisingly, that compound divergence between train and test sets is a good predictor of the test accuracy (Section 5).

## 2    DISTRIBUTION-BASED COMPOSITIONALITY ASSESSMENT (DBCA)

Like other authors, we propose to measure a learner's ability to *generalize compositionally* by using a setup where the train and test sets come from *different* distributions. More specifically, we propose a setup where each example is obtained by composing primitive elements (*atoms*), and where these atoms are similarly represented in the train and test sets while the test set contains novel *compounds*, i.e., new ways of composing the atoms of the train set.

As a simple illustrative scenario, consider the task of answering simple questions such as "Who directed Inception?" and "Did Christopher Nolan produce Goldfinger?". In this scenario, the *atoms* intuitively correspond to the primitive elements that are used to compose those questions, such as the predicates "direct(ed)" and "produce(d)", the question patterns "Who [predicate] [entity]" and "Did [entity1] [predicate] [entity2]", and the entities "Inception", "Christopher Nolan", etc. The *compounds* on the other hand correspond to the combinations of these atoms that appear in the various examples: "Who directed [entity]?", "Did Christopher Nolan [predicate] Inception?", etc.

To measure compositional generalization on such a task, one might therefore use the questions "Who directed Inception?" and "Did Christopher Nolan produce Goldfinger?" as training examples while testing on questions such as "Did Christopher Nolan direct Goldfinger?" and "Who produced Inception?" because the atoms are identically represented in the train and test sets while the compounds differ.

To make this intuition more precise, we focus on datasets such as CFQ (introduced in Section 3) and SCAN (Lake & Baroni, 2018), where each example can be created from a formal set of rules by successively applying a number of these rules. In this case, the atoms are the individual rules, while the compounds are the subgraphs of the directed acyclic graphs (DAGs) that correspond to the rule applications. (See Sections 3 and 4 for more details.)

### 2.1    PRINCIPLES FOR MEASURING COMPOSITIONALITY

We use the term *compositionality experiment* to mean a particular way of splitting the data into train and test sets with the goal of measuring compositional generalization. Based on the notions of atoms and compounds described above, we say that an ideal compositionality experiment should adhere to the following two principles:

1. *Similar atom distribution*: All atoms present in the test set are also present in the train set, and the distribution of atoms in the train set is as *similar* as possible to their distribution in the test set.

2. *Different compound distribution*: The distribution of compounds in the train set is as *different* as possible from the distribution in the test set.

The second principle guarantees that the experiment is compositionally challenging in the sense that it tests the learner on compounds that are as different as possible from the compounds used during training. The first principle aims to guarantee that the experiment is exclusively measuring the effect of the difference in the way atoms are composed to form compounds (rather than some related but different property such as domain adaptation on the distribution of the atoms).

To determine to which degree a certain experiment adheres to these principles, we use the following formalization. For a sample set $T$, we use $\mathcal{F}_A(T)$ to denote the frequency distribution of *atoms*

---

[1]Available at https://github.com/google-research/google-research/tree/master/cfq

in $T$ and $\mathcal{F}_C(T)$ for the weighted frequency distribution of *compounds* in $T$, which correspond to the *subgraphs* of the rule application DAGs. For practicality, we do not consider all subgraphs of rule application DAGs when computing the compound divergence. Instead, we first generate a large subset $\mathbb{G}$ of subgraphs, then weight them in context of their occurrence, and keep only the ones with highest sum of weights. The purpose of the weighting is to avoid double-counting compounds that are highly correlated with some of their super-compounds. We achieve this by calculating the weight of $G \in \mathbb{G}$ in a sample as $w(G) = \max_{g \in \mathrm{occ}(G)} (1 - \max_{G': g \prec g' \in \mathrm{occ}(G')} P(G'|G))$, where $\mathrm{occ}(G)$ is the set of all occurrences of $G$ in the sample, $\prec$ denotes the strict subgraph relation, and $P(G'|G)$ is the empirical probability of $G'$ occurring as a supergraph of $G$ over the full sample set. See Appendix L.4 for example subgraphs and more details on the weighting.

We measure divergence (or similarity) of the weighted distributions using the Chernoff coefficient $C_\alpha(P\|Q) = \sum_k p_k^\alpha q_k^{1-\alpha} \in [0,1]$ (Chung et al., 1989). For the atom divergence, we use $\alpha = 0.5$, which corresponds to the Bhattacharyya coefficient and reflects the desire of making the atom distributions in train and test as similar as possible. For the compound divergence, we use $\alpha = 0.1$, which reflects the intuition that it is more important whether a certain compound occurs in $P$ (train) than whether the probabilities in $P$ (train) and $Q$ (test) match exactly. This allows us to formally define as follows the notions of compound divergence $\mathcal{D}_C$ and atom divergence $\mathcal{D}_A$ of a compositionality experiment consisting of a train set $V$ and a test set $W$:

$$\mathcal{D}_C(V\|W) = 1 - C_{0.1}(\mathcal{F}_C(V)\,\|\,\mathcal{F}_C(W))$$
$$\mathcal{D}_A(V\|W) = 1 - C_{0.5}(\mathcal{F}_A(V)\,\|\,\mathcal{F}_A(W))$$

Based on these principles, we suggest to use as a preferred compositionality benchmark for a given dataset the accuracy obtained by a learner on splits with maximum compound divergence and low atom divergence (we use $\mathcal{D}_A \leq 0.02$). See Section 4 for details about how to construct such splits.

## 3 THE CFQ DATASET

We present the *Compositional Freebase Questions (CFQ)* as an example of how to construct a dataset that is specifically designed to measure compositional generalization using the DBCA method introduced above. CFQ is a simple yet realistic, large dataset of natural language questions and answers that also provides for each question a corresponding SPARQL query against the Freebase knowledge base (Bollacker et al., 2008). This means that CFQ can be used for semantic parsing (Berant et al., 2013; Yao & Van Durme, 2014), which is the task that we focus on in this paper.

### 3.1 AUTOMATIC, RULE-BASED GENERATION

Saxton et al. (2019) describe a number of benefits for automated rule-based dataset generation, including scalability, control of scope, and avoidance of human errors. Beyond these benefits, however, such an approach is particularly attractive in the context of measuring compositional generalization using the DBCA method, as it allows us to precisely track the atoms (rules) and compounds (rule applications) of each example by recording the sequence of rule applications used to generate it.

Since the way we measure compositionality depends on how the examples can be broken down into atoms and compounds, we design the generation rules so as to have few and meaningful atoms. More precisely, we aim to have as few rules as possible so that the richness of the examples comes from composing them, which yields a large variety of compounds (enabling a large range of different compound divergences) while making it easy to obtain similar distributions of atoms. Also, we aim to make our rules truly "atomic" in the sense that the behavior of any rule is independent of the context where it is applied (e.g., rules may not contain "if-then-else" constructs).

In order to minimize the number of rules, we use an intermediate *logical form* that serves as a uniform semantic representation with relatively direct mappings to natural language and SPARQL. Our rules thus fall into the following four categories (a selection of rules is provided in Appendix M):

1. *Grammar rules* that generate natural language constructs and corresponding logical forms.
2. *Inference rules* that describe transformations on logical forms, allowing us to factor out transformations that are independent of specific linguistic and SPARQL constructs.
3. *Resolution rules* that map constructs of the logical form to SPARQL constructs.

Figure 1: Generating a natural language question together with its SPARQL query using four types of rules. One (of potentially many) intermediate logical forms is also shown.

4. *Knowledge rules* that supply logical form expressions that are universally applicable. Other rules can be kept more generic by parameterizing them on knowledge.

These rules define a language of triples of the form ⟨question, logical form, SPARQL query⟩. Our generation algorithm produces such triples in a mixed top-down and bottom-up fashion. We first apply grammar rules and inference rules to produce the natural language questions and their semantics in our logical form. Then we apply resolution rules to obtain the SPARQL query. See Figure 1 for an illustration. In addition, the generator produces a normalized, directed acyclic graph (DAG) of rule applications that corresponds to the normalized program that generated the triple. (Appendix L shows an example.) Edges of this DAG represent dependencies among the rule applications, and the normalization ensures that a certain rule combination is represented using the same DAG across all the examples where it occurs.

The described approach can generate a potentially infinite set of questions, from which we first sample randomly and then subsample (to maximize the overall diversity of rule combinations while keeping a uniform distribution over complexity). We measure the diversity of rule combinations using the empirical entropy of a weighted subset of the rule application DAGs, and we use the number of rule applications as a measure of the complexity of an example. We also limit the maximum example complexity such that the questions remain relatively natural. Table 1 shows examples of generated questions at varying levels of complexity. An example of a complete data item is shown in Appendix A, a more detailed data quality analysis is presented in Appendix B, and the generation algorithm is discussed in more detail in Appendix K.

## 3.2 DATASET DETAILS AND STATISTICS

**Input and output.** While the primary focus of the dataset is semantic parsing (natural language question to SPARQL query), we also provide natural language answers for each question. This allows the dataset to be used in a text-in-text-out scenario as well (see Appendix A).

**Ambiguity.** We largely avoid ambiguity in the questions. In particular, we make sure each name is used to refer to exactly one entity, and we avoid different possible parse trees, different interpretations of plurals, and the need for disambiguation that requires semantic knowledge.

**Scope.** We select the following language features as compositional building blocks: open questions and closed questions; subordinate clauses; active and passive voice; conjunctions of verb phrases and of noun phrases; possessives with roles ("X's parent"); adjectives; and type restrictions. For knowledge base features, we select roles, verbs, types, and adjectives from domains that are well-represented in Freebase and that can be combined easily. We start from the popular movie do-

Table 1: Examples of generated questions at varying levels (L) of complexity.

| L | Question ↦ Answer |
|---|---|
| 10 | What did [Commerzbank] acquire? ↦ Eurohypo; Dresdner Bank |
| 15 | Did [Dianna Rhodes]'s spouse produce [Soldier Blue]? ↦ No |
| 20 | Which costume designer of [E.T.] married [Mannequin]'s cinematographer? ↦ Deborah Lynn Scott |
| 30 | Who was influenced by and influenced [Steve Vai], [Marx Brothers], [Woody Allen], and [Steve Martin]? ↦ Brendon Small |
| 40 | Was [Weekend Cowgirls] produced, directed, and written by a film editor that [The Evergreen State College] and [Fairway Pictures] employed? ↦ No |
| 50 | Were [It's Not About the Shawerma], [The Fifth Wall], [Rick's Canoe], [White Stork Is Coming], and [Blues for the Avatar] executive produced, edited, directed, and written by a screenwriter's parent? ↦ Yes |

Table 2: (a) CFQ dataset statistics. (b) CFQ complexity statistics in comparison to other semantic parsing datasets. Datasets in the first section map text to SQL for various DBs, with numbers as reported by Finegan-Dollak et al. (2018). Datasets in the second section map text to SPARQL for Freebase. The number of query patterns is determined by anonymizing entities and properties.

| (a) | | | (b) | | | Query |
|---|---|---|---|---|---|---|
| CFQ Dataset Statistics | | Value | Dataset | Questions | Queries | patterns |
| Unique questions | | 239,357 | Academic | 196 | 185 | 92 |
| Question patterns (mod entities) | | 239,357 | Advising | 4,570 | 211 | 174 |
| Question patterns (mod entities, verbs, etc.) | | 49,320 | AMDB | 131 | 89 | 52 |
| Unique queries | | 228,149 | ATIS | 5,280 | 947 | 751 |
| Query patterns (mod entities) | | 123,262 | GeoQuery | 877 | 246 | 98 |
| Query patterns (mod entities and properties) | | 34,921 | Restaurants | 378 | 23 | 17 |
| Open questions | | 108,786 | Scholar | 817 | 193 | 146 |
| Closed questions (with answer "yes") | | 65,092 | WikiSQL | 80,654 | 77,840 | 488 |
| Closed questions (with answer "no") | | 65,479 | Yelp | 128 | 110 | 89 |
| | | | WebQuestionsSP | 4,737 | 3,750 | 240 |
| | | | ComplexWebQuestions | 34,689 | 27,875 | 2,474 |
| | | | **CFQ (this work)** | **239,357** | **228,149** | **34,921** |

main (e.g., directing, producing, editor, sequel) and extend this with personal relations (e.g., parent, spouse, sibling), companies (e.g., founding, employer), and adjectives (e.g., gender, nationality).

**Logical form and grammar.** For the internal logical form, we adopt a variation of the description logic $\mathcal{EL}$ (Baader et al., 2003; 2005), augmented with additional constructors (see Appendix I) to more easily map to certain linguistic structures. For the grammar rules, we use a unification-based grammar syntax similar to that used in the Prolog extension GULP 3.1 (Covington, 1994), with addition of support for disjunction, negation, absence, and default inheritance of features for compactness of representation.

**Grounding in Freebase.** Once an example is generated by the CFQ rules, it still contains entity placeholders instead of Freebase machine ids (MIDs). For the task of semantic parsing, the examples could theoretically be used as-is, as our avoidance of semantic ambiguity means that a learner should not need knowledge of the specific entity in order to parse the question. To make the questions natural, however, we apply an additional step of replacing the placeholders with appropriate specific entities. To do this we first execute the generated SPARQL query against Freebase. This returns a set of candidate MID combinations that satisfy the query and can be used as substitutes. If the set is empty, we abandon the generated question candidate as unnatural. Otherwise, we pick one combination at random to yield a question with positive answer. In the case of a closed question, we also generate a variation that yields the answer "No", which we do by mixing in MIDs from another substitution (or a more generic replacement if that fails) to keep the question as plausible-sounding as possible. We then randomly choose either the question with positive or with negative answer, to avoid spurious correlations between question structure and yes/no answer.

**Semantic and structural filtering.** Even among the questions that can be satisfied in Freebase, there are some that are meaningful but somewhat unnatural, such as "Was Strange Days directed by a female person whose gender is female?". We automatically filter out such unnatural questions using semantic and structural rules. Note that since we do not require a learner to identify such questions, we do not track these filtering rules.

**Release and statistics.** CFQ contains 239,357 English question-answer pairs that are answerable using the public Freebase data. (The data URL is not yet provided for anonymous review.) We include a list of MIDs such that their English names map unambiguously to a MID. Table 2(a) summarizes the overall statistics of CFQ. Table 2(b) uses numbers from (Finegan-Dollak et al., 2018) and from an analysis of WebQuestionsSP (Yih et al., 2016) and ComplexWebQuestions (Talmor & Berant, 2018) to compare three key statistics of CFQ to other semantic parsing datasets (none of which provide annotations of their compositional structure). CFQ contains the most query patterns by an order of magnitude and also contains significantly more queries and questions than the other datasets. Note that it would be easy to boost the raw number of questions in CFQ almost arbitrarily by repeating the same question pattern with varying entities, but we use at most one entity substitution per question pattern. Appendix C contains more detailed analyses of the data distribution.

Table 3: Comparison of relevant measurements for different split methods on CFQ / SCAN.

| | Split Method | $\mathcal{D}_A$ Atom Divergence | $\mathcal{D}_C$ Compound Divergence | Output Pattern Coverage | Input Pattern Coverage | Output Length Ratio | Input Length Ratio |
|---|---|---|---|---|---|---|---|
| CFQ | Random | 0.000 | 0.000 | 0.726 | 0.705 | 1.007 | 1.003 |
| | Output Length | 0.033 | 0.176 | 0.000 | 0.004 | 0.486 | 0.648 |
| | Input Length | 0.047 | 0.062 | 0.285 | 0.047 | 0.584 | 0.578 |
| | Output Pattern | 0.000 | 0.008 | 0.000 | 0.516 | 0.977 | 0.984 |
| | Input Pattern | 0.000 | 0.005 | 0.636 | 0.000 | 1.028 | 1.017 |
| | **MCD$_1$** | 0.020 | **0.694** | 0.079 | 0.032 | 0.732 | 0.871 |
| | **MCD$_2$** | 0.020 | **0.713** | 0.023 | 0.007 | 0.838 | 0.958 |
| | **MCD$_3$** | 0.020 | **0.704** | 0.034 | 0.027 | 0.807 | 0.896 |
| SCAN | Random | 0.000 | 0.047 | 1.000 | 1.000 | 0.998 | 0.994 |
| | Output Length | 0.034 | 0.437 | 0.000 | 1.000 | 0.367 | 0.856 |
| | Input Length | 0.106 | 0.380 | 0.278 | 0.000 | 0.501 | 0.771 |
| | Output Pattern | 0.003 | 0.221 | 0.000 | 0.967 | 1.081 | 0.989 |
| | Input Pattern | 0.005 | 0.240 | 0.951 | 0.000 | 0.993 | 0.967 |
| | **MCD$_1$** | 0.015 | **0.736** | 0.260 | 0.357 | 0.698 | 0.926 |
| | **MCD$_2$** | 0.020 | **0.734** | 0.259 | 0.010 | 0.757 | 0.837 |
| | **MCD$_3$** | 0.014 | **0.735** | 0.318 | 0.009 | 0.632 | 0.938 |

## 4 COMPOSITIONALITY EXPERIMENTS FOR CFQ AND SCAN

The DBCA principles described in Section 2.1 enable a generic and task-independent method for constructing compositionality experiments. To construct such an experiment for a dataset $U$ and a desired combination of atom and compound divergences, we use an iterative greedy algorithm that starts with empty sets $V$ (train) and $W$ (test), and then alternates between adding an example $u \in U$ to $V$ or $W$ (while maintaining the desired train/test ratio). At each iteration, the element $u$ is selected such that $\mathcal{D}_C(V\|W)$ and $\mathcal{D}_A(V\|W)$ are kept as closely as possible to the desired values. To reduce the risk of being stuck in a local optimum, we also allow removing examples at certain iterations.

In general, there are many different splits that satisfy a desired compound and atom divergence. This reflects the fact that a certain compound may either occur exclusively in the train set or the test set, or it may occur in both of them because the split may have achieved the desired compound divergence by separating other (possibly orthogonal) compounds. Our greedy algorithm addresses this by making random choices along the way, starting with picking the first example randomly.

For the goal of measuring compositional generalization as accurately as possible, it is particularly interesting to construct *maximum compound divergence (MCD) splits*, which aim for a maximum compound divergence at a low atom divergence (we use $\mathcal{D}_A \leq 0.02$). Table 3 compares the compound divergence $\mathcal{D}_C$ and atom divergence $\mathcal{D}_A$ of three MCD splits to a random split baseline as well as to several previously suggested compositionality experiments for both CFQ and the existing SCAN dataset (cf. Section 5.3). The split methods (beyond *random* split) are the following:

- *Output length*: Variation of the setup described by Lake & Baroni (2018) where the train set consists of examples with output (SPARQL query or action sequence) length $\leq N$, while the test set consists of examples with output length $> N$. For CFQ, we use $N = 7$ constraints. For SCAN, we use $N = 22$ actions.

- *Input length*: Variation of the above setup, in which the train set consists of examples with input (question or command) length $\leq N$, while test set consists of examples with input length $> N$. For CFQ, we use $N = 19$ grammar leaves. For SCAN, we use $N = 8$ tokens.

- *Output pattern*: Variation of setup described by Finegan-Dollak et al. (2018), in which the split is based on randomly assigning clusters of examples sharing the same output (query or action sequence) pattern. Query patterns are determined by anonymizing entities and properties; action sequence patterns collapse primitive actions and directions.

- *Input pattern*: Variation of the previous setup in which the split is based on randomly assigning clusters of examples sharing the same input (question or command) pattern. Question patterns are determined by anonymizing entity and property names; command patterns collapse verbs and the interchangeable pairs left/right, around/opposite, twice/thrice.

All of these experiments are based on the same train and validation/test sizes of 40% and 10% of the whole set, respectively. For CFQ, this corresponds to about 96k train and 12k validation and test examples, whereas for SCAN, it corresponds to about 8k train and 1k validation and test examples. We chose to use half of the full dataset for the train-test splits, as it led to an appropriate balance between high compound divergence and high train set size in informal experiments.

The MCD splits achieve a significantly higher compound divergence at a similar atom divergence when compared to the other experiments. The reason for this is that, instead of focusing on only one intuitive but rather arbitrary aspect of compositional generalization, the MCD splits aim to optimize divergence across all compounds directly.

Interestingly, the MCD splits still correlate with the aspects of compositional generalization that are targeted by the other experiments in this table. As shown in the four right columns of Table 3, for each MCD split, the train set $V$ contains on average shorter examples than the test set $W$ (measured by the ratio of average lengths), and $V$ also contains only a small fraction of the input and output patterns used in $W$ (measured by the fraction of patterns covered). However, these correlations are less pronounced than for the experiments that specifically target these aspects, and they vary significantly across the different MCD splits.

This illustrates that MCD splits are comprehensive in the sense that they cover many different aspects of compositional generalization, especially when looking at multiple of them. It also means that whether a certain example ends up in train or test is not determined solely by a single criterion that is immediately observable when looking at the input and output (such as length). As we show in Appendix D.1, this generally makes the examples in train and test look fairly similar.

## 5 EXPERIMENTAL RESULTS AND ANALYSIS

### 5.1 EXPERIMENT SETUP

We use three encoder-decoder neural architectures as baselines: (1) *LSTM+attention* as an LSTM (Hochreiter & Schmidhuber, 1997) with attention mechanism (Bahdanau et al., 2015); (2) *Transformer* (Vaswani et al., 2017) and (3) *Universal Transformer* (Dehghani et al., 2018).

We tune the hyperparameters using a CFQ *random* split, and we keep the hyperparameters fixed for both CFQ and SCAN (listed in Appendix E). In particular the number of training steps is kept constant to remove this factor of variation. We train a fresh model for each experiment, and we replicate each experiment 5 times and report the resulting mean accuracy with 95% confidence intervals.

Note that while we construct test and validation sets from the same distribution, we suggest that hyperparameter tuning should be done on a random split (or random subset of the train set) if one wants to measure compositional generalization of a *model* with respect to an *unknown* test distribution as opposed to an *architecture* with respect to a *known* test distribution. Tuning on a validation set that has the same distribution as the test set would amount to optimizing for a particular type of compound divergence and thus measure the ability for a particular architecture to yield models that can be made to generalize in one particular way (through leaking information about the test set in the hyperparameters).

Similarly to Finegan-Dollak et al. (2018), we anonymize the Freebase names and MIDs in the textual input and the SPARQL output, respectively, by replacing them with a placeholder (e.g., "M0" for the first MID). This removes the need for two learning sub-tasks that are orthogonal to our focus: named entity recognition and learning that the MIDs are patterns that need to be copied. An example input-output (question-query) pair then looks like the following: 'Was M0 a screenwriter' $\mapsto$ 'select count(*) where {M0 a ns:film.writer}'.

The main relation we are interested in is the one between compound divergence of the data split and accuracy. Specifically, we compute the accuracy of each model configuration on a series of divergence-based splits that we produce with target compound divergences that span the range between zero and the maximum achievable in 0.1 increments (while ensuring that atom divergence does not exceed the value of 0.02). For each target divergence, we produce at least 3 different splits with different randomization parameters (compare Section 4). For comparison, we also compute accuracies on the other splits shown in Table 3.

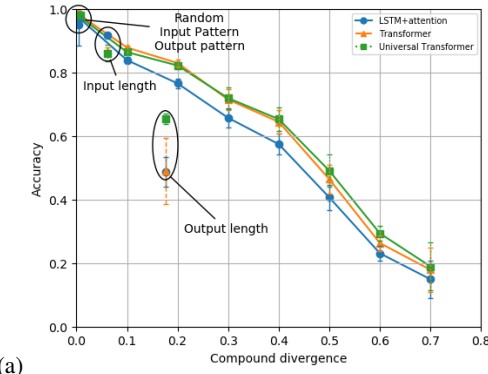 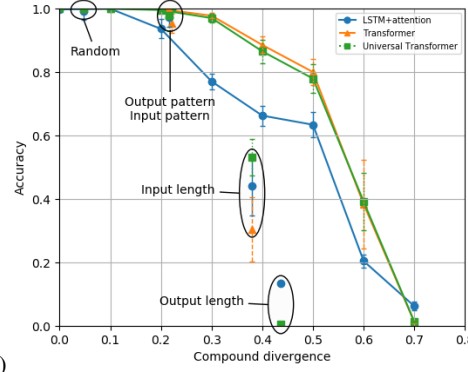

(a)                                                         (b)

Figure 2: Accuracies of the three baseline systems on (a) CFQ and (b) SCAN vs. compound divergence for different split methods and for different target compound divergences.

Table 4: Mean accuracies of the three baseline systems on CFQ and SCAN (in %).

| Dataset | CFQ | | SCAN | |
|---|---|---|---|---|
| Split Method | Random | MCD | Random | MCD |
| LSTM+attention | 97.4 ±0.3 | 14.9 ±1.1 | 99.9 ±2.7 | 6.1 ±2.2 |
| Transformer | 98.5 ±0.2 | 17.9 ±0.9 | 100.0 ±0.0 | 1.1 ±0.5 |
| Universal Transformer | 98.0 ±0.3 | 18.9 ±1.4 | 99.9 ±0.2 | 1.2 ±0.7 |

## 5.2 RESULTS AND ANALYSIS FOR CFQ

The mean accuracies of the three architectures on CFQ are shown in Figure 2(a) and Table 4. We make three main observations:

- All models achieve an accuracy larger than 95% on a random split, and this is true even if they are trained on 10 times fewer training instances (see Appendix H for a more detailed analysis on the performance with varying training size).

- The mean accuracy on the MCD splits is below 20% for all architectures, which means that even a large train set (about 96k instances) with a similar distribution of atoms between train and test is not sufficient for these architectures to perform well on the test distribution.

- For all architectures, there is a strong negative correlation between the compound divergence and the mean accuracy.

This suggests that the baseline models are able to capture the superficial structure of the dataset, but fail to capture the compositional structure. We find it surprising that varying the compound divergence gives direct control of the (mean) accuracy, even though the examples in train and test look similar (see Appendix D.1). This means that compound divergence seems to capture the core difficulty for these ML architectures to generalize compositionally.

Note that the experiment based on output-length exhibits a worse accuracy than what we would expect based on its compositional divergence. One explanation for this is that the test distribution varies from the training distribution in other ways than compound divergence (namely in output length and a slightly higher atom divergence), which seems to make this split particularly difficult for the baseline architectures. To analyze the influence of the length ratio further, we compute the correlation between length ratios and accuracy of the baseline systems and compare it to the correlation between compound divergence and accuracy. We observe $R^2$ correlation coefficients between 0.11 and 0.22 for the input and output length ratios and between 0.81 and 0.88 for the compound divergence. This shows that despite the known phenomenon that the baseline systems struggle to generalize to longer lengths, the compound divergence seems to be a stronger explanation for the accuracy on different splits than the lengths ratios.

**Error analysis.** We perform an analysis of the errors for the split $MCD_1$ (the first MCD split that we constructed, with more details provided in Appendix F). We observe accuracies between 29%

and 37% on the test set of this particular split. Qualitatively, all three systems seem to make similar errors at this point (68% of errors are on the same samples). They make more errors for longer sequences and predict about 20% too short output when they make an error. The most common category of error is the omission of a clause in the output (present in 43%-49% of the test samples), e.g.: (1) Omitted conjunctions: for the input "What spouse of a film producer executive produced and edited M0, M1, and M2?" the best system ignores "executive produced" in the output. (2) Omitted adjectives: for the input "Which female Spanish film producer was M3' s spouse?" the best system ignores the adjective "female".

## 5.3 RESULTS AND ANALYSIS FOR SCAN

To demonstrate the use of our analysis method on another dataset, we re-create the SCAN data-set (Lake & Baroni, 2018), which consists of compositional navigation *commands* (e.g, 'turn left twice and jump') mapped to corresponding *action sequences* (e.g., 'LTURN LTURN JUMP'). We use the original grammar while tracking the rule applications used for the construction of each input-output pair. This enables us to compare the compositional generalization abilities of the baseline systems on this dataset in a novel way.

Figure 2(b) shows the graphs for the SCAN data set in the same setup as Figure 2(a) does for CFQ. We observe that the compound divergence again is a good predictor for the mean accuracy for all three architectures. One difference is that for SCAN the systems are able to attain accuracies close to 100% for compound divergences up to around 0.2, which is not the case for CFQ. This seems to be in line with the fact that overall CFQ is a more complex task than SCAN: the total number of rules used in generating SCAN is only 38 in comparison to 443 rules in the construction of CFQ.

Appendix G provides a comparison to other experiments presented in previous work, including experiments that have significantly different atom distributions. We observe that this generally causes lower accuracies but does not break the correlation between accuracy and compound divergence.

## 6 RELATED WORK

To measure compositional generalization for semantic parsing to SQL, Finegan-Dollak et al. (2018) propose to ensure that no SQL query pattern occurs in both the train and the test set ("query split"), and they provide such splits for several data sets. By evaluating several ML architectures the authors confirm that this query-pattern split is harder to learn than a conventional split.

Lake & Baroni (2018) introduce the SCAN dataset, and several publications provide interesting analyses of compositional generalization using it (Bastings et al., 2018; Loula et al., 2018). Russin et al. (2019) discuss a particular extension of a seq2seq model that is effective in handling difficult SCAN sub-tasks by separating semantic and syntactic information during learning. Our contributions extend the analyses on the SCAN data in several ways: CFQ provides richer annotations and covers a broader subset of English than the SCAN dataset, and we propose a comprehensive score for assessing aggregate compositionality of a system on a given task.

The mathematics dataset (Saxton et al., 2019) is a large, automatically generated set of 112M samples in 56 separated sub-tasks. The authors present data and experiments that share common goals with our approach, but focus on mathematical reasoning instead of natural language. Our breakdown of generation rules per train sample is more fine-grained, which allows a more precise compositional generalization analysis. Being automatically generated also links our approach to datasets such as the bAbI tasks (Weston et al., 2016), which however do not focus on compositional generalization.

A dataset related to CFQ is ComplexWebQuestions (Talmor & Berant, 2018), which consists of complex questions that are automatically generated from simpler sub-questions in WebQuestionsSP (Yih et al., 2016) and then reworded manually. While these datasets can be used for semantic parsing, we did not find them suitable for a thorough compositionality analysis because a consistent annotation with the compositional structure would be hard to obtain. Other approaches to semi-automatic dataset creation also use paraphrasing (Wang et al., 2015; Su et al., 2016).

Johnson et al. (2017) introduce the generated CLEVR dataset, which shares common goals with our work applied in the area of visual reasoning. The dataset's functional programs capture some of the structural information of the questions and are linked one-to-many to the 423 question patterns

used. The authors specifically investigate generalization to new combinations of visual attributes in one experiment which uses a particular train-test split based on the colors used. Mao et al. (2019) propose a neural-symbolic architecture and discuss promising results on additional specific splits of the CLEVR data, e.g. based on object counts and program depth. Hudson & Manning (2018) describe how the application of compositional attention networks to the CLEVR data leads to structured and data-efficient learning. Hudson & Manning (2019a) present a large, compositional, generated visual question answering data set with functional programs, on which neural state machines achieve good performance (Hudson & Manning, 2019b). The use of specific *splits* between train and test data also occurs in the context of visual data. E.g., Agrawal et al. (2018) propose a greedy split algorithm to maximize the coverage of test concepts in the train set while keeping question-type/answer pairs disjoint and observe performance degradation of existing approaches. Bahdanau et al. (2019) introduce a synthetic visual question answering dataset called SQOOP, which is used to test whether a learner can answer questions about all possible object pairs after being trained on a subset.

While these datasets are very interesting, the additional annotation that we provide in CFQ indicating the exact rule trees needed to link input and output makes additional analyses regarding compositionality possible. Our analyses go beyond many of the presented discussions (that mostly focus on accuracy regarding particular holdouts) in formalizing an approach that uses the atom and compound divergences to measure compositionality.

A number of ML approaches have been developed for semantic parsing. Miller et al. (2016) propose Key-Value Memory Networks – neural network-based architectures that internalize a knowledge base into the network – and introduce the WikiMovies dataset. Zhang et al. (2018) develop an end-to-end architecture that can handle noise in questions and learn multi-hop reasoning simultaneously. They introduce the MetaQA benchmark that is based on WikiMovies but uses a set of only 511 question patterns (mod entities) shared between train and test.

With regards to studying compositionality in ML, Battaglia et al. (2018) argue that combinatorial generalization should be a top priority to achieve human-like abilities. Andreas (2019) discusses measuring the compositionality of a trained representation, e.g. of a learned embedding. The author suggests to use a tree reconstruction error that is based on how well the oracle derivation of the input matches the structure that can be derived on the representations. Higgins et al. (2018) discuss an architecture that enables the learning of compositional concept operators on top of learned visual abstractions. Chang et al. (2019) introduce the compositional recursive learner that "can generalize to more complex problems than the learner has previously encountered".

# 7 CONCLUSION AND OUTLOOK

In this paper we presented what is (to the best of our knowledge) the largest and most comprehensive benchmark for compositional generalization on a realistic NLU task. It is based on a new dataset generated via a principled rule-based approach and a new method of splitting the dataset by optimizing the divergence of atom and compound distributions between train and test sets. The performance of three baselines indicates that in a simple but realistic NLU scenario, state-of-the-art learning systems fail to generalize compositionally even if they are provided with large amounts of training data and that the mean accuracy is strongly correlated with the compound divergence.

We hope our work will inspire others to use this benchmark as a yardstick to advance the compositional generalization capabilities of learning systems and achieve high accuracy at high compound divergence. Some specific directions that we consider promising include applying unsupervised pre-training on the input language or output queries and the use of more diverse or more targeted learning architectures, such as syntactic attention (Russin et al., 2019). We also believe it would be interesting to apply the DBCA approach to other domains such as visual reasoning, e.g. based on CLEVR (Johnson et al., 2017).

In the area of compositionality benchmarks, we are interested in determining the performance of current architectures on the end-to-end task that expects a natural language answer given a natural language question in CFQ. We would like also to extend our approach to broader subsets of language understanding, including use of ambiguous constructs, negations, quantification, comparatives, additional languages, and other vertical domains.

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

## APPENDIX

## A EXAMPLE DATASET ITEM

The following shows an example data item including the question text in various forms, the answer, the SPARQL query in various forms, some tracked statistics, and the set of used rules (atoms) and the applied rule tree (compound). Some details are omitted, indicated by ellipses ('...').

```
"question": "Did Agustin Almodovar executive produce Deadfall",
"questionWithBrackets": "Did [Agustin Almodovar] executive produce [Deadfall]",
"questionWithMids": "Did m.04lhs01 executive produce m.0gx0plf",
"questionPatternModEntities": "Did M0 executive produce M1",
"questionTemplate": "Did [entity] [VP_SIMPLE] [entity]",
"expectedResponse": "No",
"sparql": "SELECT count(*) WHERE {\nns:m.04lhs01 ns:film.producer.films_executive_produced ns:
    m.0gx0plf\n}",
"sparqlPatternModEntities": "SELECT count(*) WHERE {\nM0 ns:film.producer.
    films_executive_produced M1\n}",
"sparqlPattern": "SELECT count(*) WHERE {\nM0 P0 M1\n}",
"complexityMeasures": {
 "parseTreeLeafCount": 5,
 "parseTreeRuleCount": 12
 "sparqlMaximumChainLength": 2,
 "sparqlMaximumDegree": 1,
 "sparqlNumConstraints": 1,
 "sparqlNumVariables": 0,
},
"aggregatedRuleInfo": {
  "ruleId": [
  {
   "type": "SPARQL_GENERATION",
   "stringValue": "ENTITY_MID"
  },
  {
   "type": "SPARQL_GENERATION",
   "stringValue": "GET_SET_TRUTH"
  },
  {
   "type": "KNOWLEDGE",
   "stringValue": "FreebasePropertyMapping(RolePair(Executive producer, Executive producee), '
       ns:film.producer.films_executive_produced')"
  },
  {
   "type": "GRAMMAR_RULE",
   "stringValue": "YNQ=DID_DP_VP_INDIRECT"
  },
  {
   "type": "GRAMMAR_RULE",
   "stringValue": "ACTIVE_VP=VP_SIMPLE"
  },
  ...
 ],
},
"ruleTree": {
 "ruleId": {
  "type": "SPARQL_GENERATION",
  "stringValue": "CONCEPT_TO_SPARQL"
 },
 "subTree": [
  {
   "ruleId": {
    "type": "GRAMMAR_RULE",
    "stringValue": "S=YNQ"
   },
   "subTree": [
    {
     "ruleId": {
      "type": "GRAMMAR_RULE",
     "stringValue": "YNQ=DID_DP_VP_INDIRECT"
...
```

## B    DATA QUALITY ANALYSIS

During the development of our data generation pipeline, we manually checked the generated examples for quality. Below is a random selection of 50 examples of the final CFQ dataset (no cherry-picking was used). Brackets around [entity names] are provided just for ease of human reading. Manual checking also indicated that all questions are associated with the semantically correct SPARQL queries. However, because we rely on the data present in Freebase, there are three debatable questions which sound somewhat unnatural (3, 21, and 29, see further discussion below the list).

1. Who was a writer, star, and cinematographer of [Tetsuo: The Bullet Man], [Nightmare Detective], and [Bullet Ballet]?

2. Which male person was a sibling of [Andrew Klavan]?

3. Did [Wallace Stevens] influence [Levi Seeley]'s spouse and parent?

4. Did a producer, writer, and art director of [Thelma & Luis] produce, direct, and write [Light Girls]?

5. Were [Hangover Square], [Zack and Miri Make a Porno], and [Clerks II] edited by a founder and employee of a film producer?

6. What American parent of [Charlie Sistovaris] was a British screenwriter's sibling?

7. Did [Anne Williams Rubinstein] marry a person that influenced a screenwriter and influenced [John Most]?

8. Was [Cachún cachún ra ra!]'s director a film director's American child?

9. Did [Maisy's Garden]'s executive producer write, edit, and executive produce [Pakalppooram], [It's Not About the Shawerma], [Rick's Canoe], and [The Fifth Wall]?

10. Was [Holly Ellenson]'s child [Wally Ellenson]?

11. Did [Emerald Cities]'s cinematographer, writer, and editor edit, executive produce, and direct [Blues for the Avatar] and [White Stork Is Coming]?

12. Was a film producer [Lilies of the Ghetto]'s distributor and producer?

13. Which child of [Mimi Iger] did a film producer employ and [The Walt Disney Company] employ?

14. What Japanese spouse of [Hong Kong Paradise]'s star did [Ineko Arima] and [Nishiki Kô] marry?

15. Who influenced and was influenced by [Black Dynamite]'s star?

16. What was written by, edited by, directed by, produced by, and executive produced by [Pauline Collins]'s child's sibling?

17. Which Swedish film director that [Théo Van Horn]'s actor influenced did [Egen ingång] star?

18. Who was influenced by [Golden Yeggs]'s star, was influenced by [Richard Pryor], was influenced by [Bill Murray], and married [Elaine Chappelle]?

19. What did [This Is My Show]'s director, cinematographer, and star direct, edit, produce, and executive produce?

20. Who was a male costume designer and director of [Ene... due... like... fake...] and [The Windmill Bar]?

21. Was [Kumudu Munasinghe] a Dutch film producer's country of nationality's employee?

22. Did an art director, editor, director, writer, cinematographer, and star of [Tetsuo II: Body Hammer] produce [Nightmare Detective], [Tetsuo: The Iron Man], and [A Snake of June]?

23. Was [Alexandra Naoum] [Monsieur Verdoux]'s producer, writer, and star?

24. What film director founded [THX], was employed by [American Zoetrope], [LucasArts], [Skywalker Sound], and [Lucasfilm], and founded [Industrial Light & Magic]?

25. What male employee of [Weta Workshop] was [Bad Taste]'s editor?

26. Were [Weta Digital] and [Weta Workshop] founded by a cinematographer and founded by a film editor?

27. What art director influenced [DreamWorks Animation]'s founder?

28. Did [Daisies] star [Fruit of Paradise]'s costume designer and writer, star [Jaromír Vomácka], and star [Jirina Myskova]?

29. What character was influenced by a costume designer, influenced by [Pedro Calderón de la Barca], influenced by [William Shakespeare] and [Luis Buñuel], and influenced by [Miguel de Unamuno]?

30. What British costume designer of [The Love Letter] and [The Chamber] was a screenwriter's child?

31. Was [Eric Massa] a cinematographer's parent's sibling's American sibling?

32. What art director of [Stepping Sisters 1932] was a parent of [Imre Sándorházi]?

33. What was executive produced by, written by, produced by, and edited by a director of [V/H/S/2]'s sequel?

34. What did an editor and cinematographer of [Tongue Twister Variations] direct?

35. Who was a Canadian screenwriter that produced [Her Painted Hero] and [The Nick of Time Baby]?

36. Which American parent of [Janet Friedman] did [Rose Friedman] influence and marry?

37. Did [George Carlin] influence [Louis C.K.: Shameless]'s executive producer and influence [Joan Rivers]?

38. Who was a male writer, star, director, and costume designer of [The Wizard of Speed and Time]?

39. Who was [Lost Boys: The Thirst]'s prequel's sequel's art director?

40. Did a cinematographer's female parent executive produce, direct, and write [Hit Dat Shit 5]?

41. Who married [Siri von Essen], influenced [A Lesson in Love]'s director and art director, influenced [Tennessee Williams], and influenced [Maxim Gorky]?

42. What Italian film director directed [Children of Hannibal]?

43. What film producer directed, wrote, edited, and produced [la estrella], [la ardilla], and [el valiente]?

44. Were [Flames: The Movie] and [Soltera] directed by a male person and executive produced by [Hilda Russoff]'s spouse?

45. Was a sibling of [Fawwaz bin Abdulaziz Al Saud] [Badr bin Abdulaziz Al Saud]'s sibling?

46. What did a sibling of [Louise Rohr] executive produce, produce, and edit?

47. Did a French cinematographer of [Le Volcan interdit] edit [The Last Bolshevik] and direct [A.K.] and [Statues Also Die]?

48. Was [Mannai Thottu Kumbidanum] directed by and written by a Dutch male cinematographer?

49. Was a director, art director, executive producer, and costume designer of [But I'm a Genderqueer] [Lauren Soldano]?

50. Was [When We Were Kings] produced by a film editor whose spouse was employed by [Royal Academy of Dramatic Art] and distributed by [PolyGram Filmed Entertainment]?

Further discussion of the debatable questions:

3. *Did [Wallace Stevens] influence [Levi Seeley]'s spouse and parent?*
   The occurrence of the seemingly implausible combination of roles "spouse and parent" is due to incorrect data in Freebase, in which there are 502 entities asserted to be both the spouse and parent of other entities. For instance, "Anne Dacre" is both the spouse and parent of "Christopher Conyers". We can also find occasional occurrences in CFQ of other implausible role combinations, such as "parent and child", "spouse and sibling" etc., triggered by similar Freebase data issues.

Table 5: Most frequent answers in CFQ.

| Frequency | Answer | Frequency | Answer |
|---|---|---|---|
| 65479 | No | 594 | David Lynch |
| 65092 | Yes | 572 | Richard Branson |
| 1581 | Promises Written in Water | 551 | Paris, je t'aime |
| 1037 | Shinya Tsukamoto | 547 | Visions of Europe |
| 907 | Vincent Gallo | 484 | Woody Allen |
| 893 | Metro-Goldwyn-Mayer | 476 | Charlie Chaplin |
| 885 | Agnès Varda | 409 | New York, I Love You |
| 858 | George Lucas | 396 | Universal Studios |
| 800 | Jacques Demy | 392 | Robert Santos |
| 742 | The ABCs of Death | 386 | Andy Warhol |
| 709 | Rick Schmidt | 385 | Georges Méliès |
| 666 | Ingmar Bergman | 370 | Chris Rock |
| 649 | Garret Schuelke | 369 | Steven Spielberg |
| 622 | Walt Disney | 361 | Jackie Chan |
| 603 | Darren Aronofsky | 356 | Orson Welles |

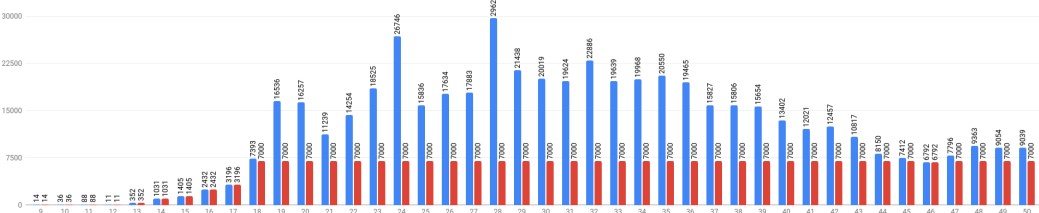

Figure 3: Number of questions by complexity before (blue) and after (red) subsampling.

21. *Was [Kumudu Munasinghe] a Dutch film producer's country of nationality's employee?*
    The somewhat unnatural phrasing of "country's employee" occurs due to a modeling choice in Freebase, in which the same entity is used to represent both a country and the government of that country. This makes it possible for a country to employ a person.

29. *What character was influenced by a costume designer, influenced by [Pedro Calderón de la Barca], influenced by [William Shakespeare] and [Luis Buñuel], and influenced by [Miguel de Unamuno]?*
    The somewhat unnatural phrasing of "a character was influenced by" occurs due to a modeling choice in Freebase, in which when a film character is based on a real person, Freebase commonly uses the same entity to represent both. This makes "person" and "character" exchangeable in the questions where the person is also a film character.

## C    DATA DISTRIBUTION ANALYSIS

### C.1    ANSWER FREQUENCIES

Table 5 shows the most frequently occurring answers in CFQ. Not surprisingly, after the answers "Yes" and "No", entities related in Freebase to the domain of movies have highest frequency.

### C.2    IMPACT OF SUBSAMPLING ON THE DISTRIBUTION OF COMPLEXITY LEVELS

Figure 3 illustrates how subsampling changes the distribution of questions in CFQ with different levels of complexity to become more even.

### C.3    IMPACT OF SUBSAMPLING ON THE FREQUENCY OF RULES AND RULE COMBINATIONS

Subsampling increases the frequency of rarely used rules and rule combinations and decreases the frequency of commonly used ones. For rules, this is illustrated by Figure 4 which shows the ratio of

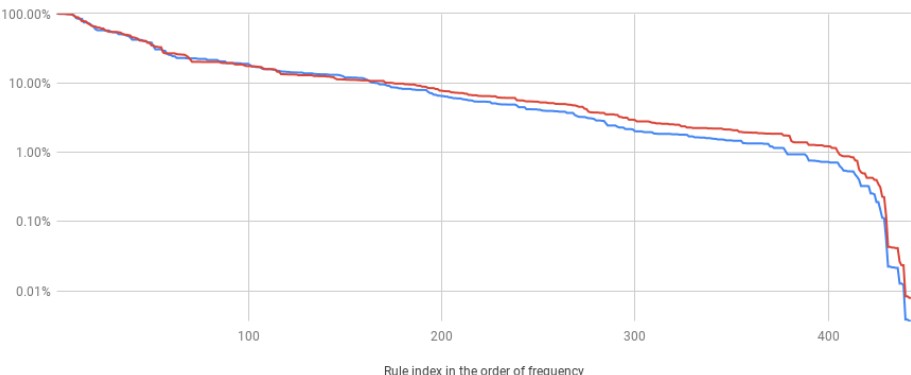

Figure 4: Ratio of examples in which a given rule appears, before (blue) and after (red) subsampling.

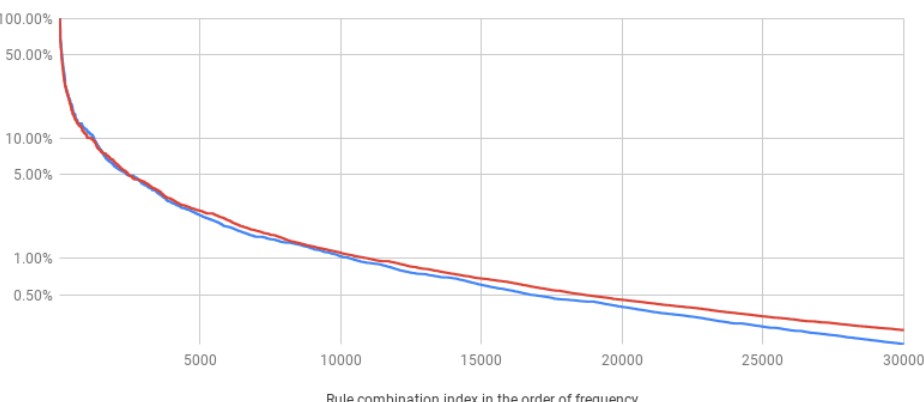

Figure 5: Ratio of examples in which a given rule combination appears, before (blue) and after (red) subsampling.

examples each rule appears in, before and after subsampling, in the order of their frequency. Figure 5 shows the same comparison for rule combinations.

## D    DIVERGENCE-BASED SPLIT ANALYSIS

### D.1    QUALITATIVE ANALYSIS OF $MCD_1$

Traditional compositionality experiments often use train-test splits based on observable properties of the input and output (e.g., input/output complexity, input/output patterns, and input/output feature holdouts). One consequence of this is that the difference between train and test examples is relatively easily observable "with the naked eye". The lists below illustrate that this is not usually the case for divergence-based splits. Similar to the random sample of the general data in Appendix B we provide a random sample of size 20 from both the train and test set here. Indeed, even for the $MCD_1$ split with a high divergence of 0.694, the 20 random samples of train and test questions shown below cannot easily be distinguished as they both contain the same kind of questions of different sizes.

Train samples from $MCD_1$:

1. What was founded by a costume designer, founded by [Forgotten Silver]'s star, and founded by [Jamie Selkirk]?

2. Which male person influenced and was influenced by [William Dean Howells]?

3. Did [Marco Bellocchio] produce, write, and direct [Greek Pete]?

4. What did [Rick Schmidt] edit, [Philip Rashkovetsky] edit, and a cinematographer edit?

5. Were [The Living Playing Cards] and [The Haunted Castle] edited by, directed by, and produced by a French writer of [Le cauchemar de Méliès]?

6. What did a spouse of [Shorts]'s producer's spouse executive produce and direct?

7. Did [P. G. Wodehouse], [Raymond Chandler], [Edward Bunker], [Pauline Kael], and [Michael Cimino] influence [Grindhouse]'s cinematographer and star?

8. What Mexican person did a film producer employ?

9. Did [The Midnight After]'s Chinese executive producer edit [Perfect Life] and [Dumplings]?

10. Who did [For the Secret Service]'s director's female spouse influence?

11. Who married, was influenced by, and influenced a company's founder?

12. Was [MAN SE]'s French male German employee's employer [Sulzer]?

13. Who influenced an actor that [Robin Santana] was influenced by and [K. J. Stevens] was influenced by and was influenced by [Virgil]?

14. Did [Pirates of Malaysia] star [Giuseppe Addobbati] and star a Spanish screenwriter?

15. Was [The Silence of the Sea] written by, produced by, executive produced by, directed by, and edited by [The Red Circle]'s French editor?

16. Did [Chanel] employ a German costume designer, employ [Gaspard Ulliel] and [Maureen Chiquet], and employ [Jacques Polge]?

17. Who was influenced by [Adam Sandler] and married a film producer?

18. Did a Spanish screenwriter's child direct and edit [Bakuchi-uchi: Nagaremono]?

19. Was a founder of [IG Port] employed by a film producer?

20. Was [Orizzonti Orizzonti!] executive produced by and written by an art director's sibling?

Test samples from MCD$_1$:

1. What sequel of [Paranormal Activity 2] was edited by and written by a film director?

2. What spouse of a film producer founded [Grand Hustle Records] and was employed by [40/40 Club], [Roc-A-Fella Records], and [Def Jam Recordings]?

3. Did [Pixar] employ an art director and employ [Susham Bedi]?

4. Was a sibling of [David Lindbland] [Dynamit Nobel]'s Swedish founder?

5. What prequel of [Charlie the Unicorn 2] starred, was edited by, was produced by, was written by, and was directed by [Jason Steele]?

6. Did [Rick Schmidt] direct, produce, executive produce, and edit [Blues for the Avatar], [White Stork Is Coming], [The Fifth Wall], and [It's Not About the Shawerma]?

7. Was [Luke Larkin Music] an art director's employer?

8. What prequel of [Goat Story 2] was executive produced, written, directed, edited, and produced by [Jan Tománek]?

9. Was [Bullet Ballet]'s editor, star, director, and cinematographer [Promises Written in Water]'s star, director, writer, executive producer, and art director?

10. What was edited by, produced by, directed by, and written by [Ellis Kaan Ozen], [Thaw Bwe], [Jeffrey Malkofsky-Berger], and [Leslie Berkley]?

11. Was a person's female sibling [Reggae in a Babylon]'s producer?

12. Who was a director, cinematographer, executive producer, art director, producer, star, and writer of [The Man Who Killed God]?

13. Was [My Sweet Home]'s director, editor, writer, art director, producer, cinematographer, and costume designer a person?

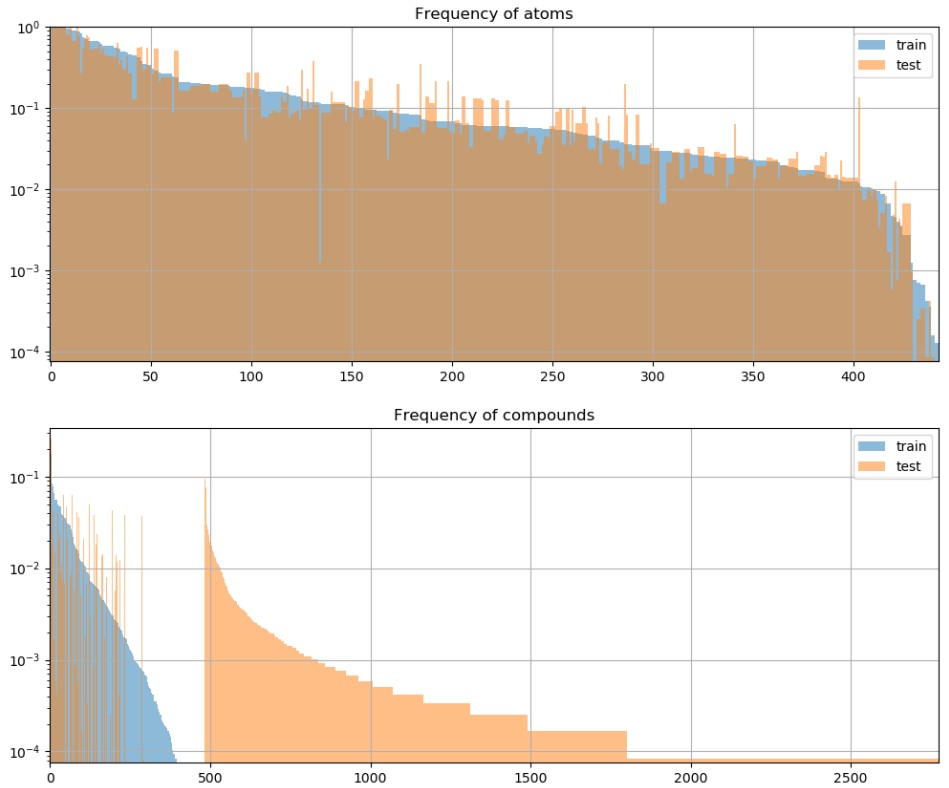

Figure 6: Frequency of atoms resp. compounds in the train vs. test set

14. Which art director, star, and editor of [The Brown Bunny] and [Promises Written in Water] did [Cord] star?

15. Did an employee and founder of [Virgin Mobile Australia], [Virgin Mobile USA], and [Virgin Mobile France] found [Virgin America] and found [V2 Records]?

16. Was a Chinese executive producer and star of [Happy Ghost II] and [All's Well, Ends Well 2010] a film director?

17. Was [The Voyeur]'s executive producer an actor's parent?

18. Did [Erasable Cities]'s writer, producer, editor, art director, cinematographer, and director produce and executive produce [Promises Written in Water]?

19. Who was an editor, star, and cinematographer of [Tetsuo: The Iron Man], [A Snake of June], and [Bullet Ballet]?

20. Was a costume designer's employer [Philips High School]?

### D.2 Quantitative analysis of $MCD_1$

Figure 6 shows the frequency of atoms (upper graph) and compounds (lower graph) in the train and test sets of the maximum compound divergence split for the CFQ data. As the frequency of an atom resp. compound we use the fraction of examples it appears in. Both atoms and compounds are indexed primarily by their frequency in the train set, secondarily by their frequency in the test set, in decreasing order. For practical reasons we only look at a small subset of compounds here but we believe the analysis is representative.

We can see that the frequency of atoms in the two sets is very aligned and that all atoms from the test set appear in the train set. The frequency of compounds however is wildly different: While some invariably occur in both sets, the frequencies are often not aligned and most compounds appear only in either the train or the test set.

Table 6: Summary of hyperparameters that deviate from the defaults. Default hyperparameter sets are: lstm_bahdanau_attention_multi, transformer_base, and universal_transformer_tiny, respectively.

|  | LSTM+attention | Transformer | Universal Transformer |
|---|---|---|---|
| train steps | 35,000 | 35,000 | 35,000 |
| batch size | 2,048 | 4,096 | 2,048 |
| hidden size | 512 | 128 | 256 |
| num hidden layers | 2 | 2 | 6 |
| num heads | – | 16 | 4 |
| learning rate schedule | – | constant*linear_warmup*rsqrt_decay | |
| learning rate {,constant} | 0.03 | 0.08 | 0.14 |
| learning rate warmup steps | – | 4,000 | 8,000 |
| dropout | 0.4 | – | – |

Table 7: Examples with a given error (in %) of total test set size. See text for details.

| System | Error type | | | | | | | | | | |
|---|---|---|---|---|---|---|---|---|---|---|---|
|  | Clause error | | | | | | Filter error | | | | Malformed output |
|  | sum | ins | del | sub | | | sum | ins | del | sub | |
|  |  |  |  | prop | node | both |  |  |  |  |  |
| LSTM+Attention | 71.7 | 9.0 | 49.2 | 23.3 | 33.1 | 27.6 | 8.6 | 2.1 | 3.2 | 4.1 | 0.2 |
| Transformer | 63.9 | 8.0 | 49.2 | 13.3 | 28.2 | 15.4 | 7.0 | 2.1 | 1.7 | 4.2 | 3.0 |
| Universal Transformer | 59.9 | 9.8 | 43.1 | 12.1 | 22.9 | 14.5 | 5.0 | 1.7 | 1.1 | 2.7 | 0.8 |

# E  HYPERPARAMETERS

The experiments were run using the tensor2tensor framework (Vaswani et al., 2018) with some of the hyperparameters tuned using a random split of a previous, smaller version of the data set during development. We use the default hyperparameter sets publicly available in the tensor2tensor implementation (obtained from https://github.com/tensorflow/tensor2tensor) and override the tuned hyperparameters. The hyperparameters used are summarized in Table 6.

# F  DETAILED ERROR ANALYSIS

## F.1  BREAKDOWN OF ERROR TYPES

Table 7 shows a more detailed analysis of the errors that the baseline models make on CFQ for $MCD_1$ (compare Section 5.2). The reported errors are bucketized into three main types: SPARQL property clause error, SPARQL filter clause error and malformed SPARQL query in the model's output. The total number of test set examples exhibiting any clause or filter error is reported (*sum* column), as well as the number of insertions (*ins*), deletions (*del*), and substitutions (*sub*) in the model's output with respect to the correct query. Property clause substitution errors are further subdivided into those where only the property itself is wrong while subject and object are correct (*prop*), those where the property is correct but either subject or object is wrong (*node*) and those where both the property and the subject or the object are wrong (*both*).

The accuracy metric requires the model response and the golden (correct) answer to be exactly equal to each other. Thus, a SPARQL query with the same clauses as the golden answer but in a different order or with some of the clauses appearing multiple times is also considered to be an error despite being equivalent to the golden answer in its meaning. The amount of such errors is relatively small though, accounting for 1.8%, 0.6% and 1.5% of total test set size for LSTM+Attention, Transformer and Universal Transformer respectively.

## F.2  QUALITATIVE ERROR ANALYSIS

Below we qualitatively analyze a number of instances the models fail on. We anonymize the MIDs in the same way as the data is provided to the models (see Section 5). We first select queries on which all machine learning systems fail in all replicated runs (about 5k instances out of a total of about

| Subquery | Count |
|---|---:|
| What. . . ? | 23,695 |
| . . . sibling of Mx . . . | 2,331 |
| . . . Mx's parent . . . | 1,222 |
| What [entity] was. . . ? | 1,066 |
| What sibling . . . ? | 0 |
| . . . [DetNP]'s [NP]. . . | 51,600 |
| . . . [NP] of [NP]. . . | 20,038 |
| What [NP] was [DetNP]? | 416 |
| What [NP] was [DetNP]'s [NP]? | 0 |

Table 8: Subqueries of "What sibling of M0 was M1' s parent?" and their occurrences in training.

12k), and then randomly select queries from this list. In the following we discuss a few cases in more detail. Note that, for readability, we use the following abbreviations for the SPARQL properties in Query 1:

- `ns:people.person.child = ns:people.person.children|`
  `ns:fictional_universe.fictional_character.children|`
  `ns:organization.organization.child/`
  `ns:organization.organization_relationship.child`

- `ns:people.person.sibling = ns:people.person.sibling_s/`
  `ns:people.sibling_relationship.sibling|`
  `ns:fictional_universe.fictional_character.siblings/`
  `ns:fictional_universe.`
  `sibling_relationship_of_fictional_characters.siblings`

*Query 1: "What sibling of M0 was M1' s parent?"*

Golden (correct) SPARQL query:

```
SELECT DISTINCT ?x0 WHERE {
  ?x0 ns:people.person.child M1 .
  ?x0 ns:people.person.sibling M0 .
  FILTER ( ?x0 != M0 )
}
```

Inferred (system) SPARQL query:

```
SELECT DISTINCT ?x0 WHERE {
  ?x0 ns:people.person.sibling ?x1 .
  ?x0 ns:people.person.sibling M0 .
  ?x1 ns:people.person.child M1 .
  FILTER ( ?x0 != ?x1 )
}
```

**Analysis.** The meaning of the SPARQL query generated by the system is "What sibling of M0 was a sibling of M1's parent?", which is incorrect. We next analyze the train set, in order to show that we believe enough information has been provided in the train set for the question to be answered correctly.

Some subqueries of the query and their occurrences are shown in Table 8. While the exact subquery "What sibling" does not occur at training, the two words have been shown separately in many instances: the subqueries "sibling of Mx", and "Mx's parent" occur 2,331 and 1,222 times, respectively. We can analyze this example in more detail by comparing parts of the rule tree of this example with those shown at training. As can be read from the table, similar sentences have been shown during training. Some examples are:

- What was executive produced by and written by a sibling of M0?

| Subquery | Count |
|---|---:|
| …direct… | 40,616 |
| …edit… | 31,776 |
| Did…? | 17,464 |
| …film director … | 5,603 |
| …male film director … | 121 |
| ……edit and direct … | 93 |
| Did [DetNP] [VP] [DetNP]? | 17,464 |
| Did [DetNP] [VP] and [VP] [DetNP]? | 1,432 |
| Did [DetNP] [VP] [DetNP] and [DetNP]? | 909 |
| Did [DetNP] [VP] and [VP] [DetNP] and [DetNP]? | 0 |

Table 9: Subqueries of "Did a male film director edit and direct M0 and M1?" and their occurrences in training.

- What costume designer did M1's parent employ?
- What cinematographer was a film editor that M2 and M3 married?
- What film director was a character influenced by M2?

*Query 2: "Did a male film director edit and direct M0 and M1?"*

Golden (correct) SPARQL query:

```
SELECT count ( * ) WHERE {
  ?x0 ns:film.director.film M0 .
  ?x0 ns:film.director.film M1 .
  ?x0 ns:film.editor.film M0 .
  ?x0 ns:film.editor.film M1 .
  ?x0 ns:people.person.gender m_05zppz
}
```

Inferred (system) SPARQL query:

```
SELECT count ( * ) WHERE {
  ?x0 ns:film.director.film M0 .
  ?x0 ns:film.director.film M1 .
  ?x0 ns:film.editor.film M0 .
  ?x0 ns:people.person.gender m_05zppz
}
```

**Analysis.** The meaning of the inferred SPARQL query is "Did a male film director edit M0 and direct M0 and M1?". It thus seems the model 'forgets' to include the relation between the director and movie M1.

Looking at subqueries and their occurrence count (Table 9), we see again that various subqueries occur often during training. However, "edit and direct" have not been shown often together. When looking at the rule trees, we see that both conjunctions in the query occur often at training separately: "Did [DetNP] [VP] and [VP] [DetNP]" occurs 1,432 times, and "Did [DetNP] [VP] [Entity] and [Entity]" occurs 909 times. However, they never occur together: "Did [DetNP] [VP] and [VP] [DetNP] and [DetNP]" does not occur at training. This may be the reason why all systems fail on this example, but at the same time we believe a compositional learner should be able to generalize correctly given the training instances. Some examples are:

- Did a male film director that M3's parent married influence an art director?
- Did a film producer that played M2 edit and direct M1?
- Did a screenwriter edit and direct a sequel of M1
- Did a Chinese male film director edit M1 and M2?

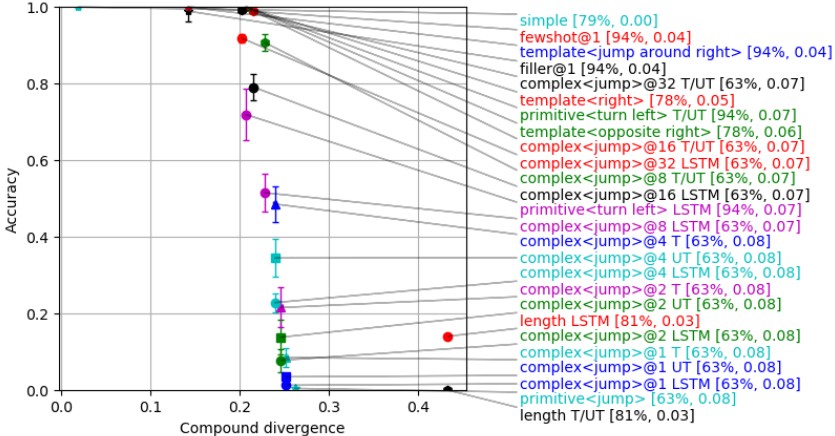

Figure 7: Accuracy and divergence measurements for splits of SCAN as used in other work (see text for details). The numbers in brackets show the train / full data-set ratio, and the atom divergence.

## G ADDITIONAL EXPERIMENTAL RESULTS ON SCAN

Figure 7 shows a scatter plot of accuracy vs. compound divergence for the three baseline architectures (see Section 5) on existing splits of the SCAN data. These splits are discussed in (Lake & Baroni, 2018) and (Loula et al., 2018), and the exact split data is available. (Data splits obtained from https://github.com/brendenlake/SCAN). We map these splits onto the re-created SCAN data, which enables us to measure the atom and compound divergences. The authors present a total of six split experiments (some with several sub-experiments):

- (Lake & Baroni, 2018):
    - *simple* (random)
    - by action sequence *length*
    - adding a *primitive* and adding a primitive along with *complex* combinations
- (Loula et al., 2018):
    - adding a *template*
    - adding template *fillers*
    - adding more training examples of fillers (*fewshot*)

In the plot, we omit some data points that are too close to be distinguished easily. The point labels have the form '(abbreviated experiment name)<(parameter)>@(number of samples) (baseline system abbreviation) [(train set size fraction), (split atom divergence)]'. The train set size fraction is given as a percentage of the overall data size. The baseline system abbreviations are LSTM, T for Transformer, UT for Universal Transformer, T/UT where both transformer models are indistinguishable, and empty where all three systems perform indistinguishably. The abbreviated experiment name is one of the names in *italics* above.

We can observe a strong dependency of the accuracies on the compound divergence of the data split. Again, this seems to indicate that the compound divergence is correlated with accuracy for these baseline architectures. One difference to the data shown in Figure 2(b) is that for this set of experiments the accuracy drops faster with increasing compound divergence. One explanation for this effect is that the experiments are directly aimed at highlighting one specific potentially problematic scenario for learning. E.g. in the experiment 'primitive<jump>' (with very low accuracies for all three systems) the jump command is shown exactly in one combination (namely alone) in the training data while it occurs in all test examples in arbitrary combinations.

This is reflected in the higher atom divergence value of 0.08 for this split, as well as in all other splits that exhibit a low accuracy at a low compound divergence in Figure 7. Note that Lake & Baroni (2018) already compare the experiment 'primitive<jump>' to the experiment 'primitive<turn left>' for which all three systems achieve a much higher accuracy. In their interpretation of this phenomenon, they mainly focus on the fact that in contrast to 'jump', the action 'turn left' is also generated by other inputs. We additionally observe that the latter experiment also has a slightly lower atom divergence of 0.07, a lower compound divergence, and it covers a much larger part of the data in the train set (94% vs. 63%).

While the accuracies we observe for the 'primitive' experiments are very much in line with the results reported by Lake & Baroni (2018), we noticed a few interesting differences for other experiments: All three systems go to 100% accuracy on the fewshot task even for one example (while Loula et al. (2018) report a slowly increasing accuracy for the architecture they evaluate). On the other hand, both transformer models only reach 0% accuracy on the length split, while the LSTM obtains around 14% (which is in line with what previous work reports).

## H   ANALYSIS OF RELATIONS BETWEEN ACCURACY, COMPOUND DIVERGENCE, AND TRAINING SIZE

Figure 2 shows for all baseline systems a strong correlation between accuracy and compound divergence for the chosen training sizes (96k for CFQ and 8k for SCAN). One interesting question is whether and how this correlation is changed for different training sizes. Figures 8 and 9 show that this correlation holds also for smaller training sizes but that the accuracy is generally somewhat lower for smaller training sizes.

At the same time, we observe that the difference between accuracies of various training sizes gets smaller as the training size increases. This can be seen even more clearly in Figures 10 and 11, which plot the training size rather than the compound divergence on the x-axis. These figures show that the increase in accuracy flattens out significantly as we reach training size of about 80k for CFQ and about 6k for SCAN. This indicates that further increasing train set size may not be sufficient to do well on these compositionality experiments.

## I   LOGICAL FORM

To represent our logical form we use syntax of the description logic $\mathcal{EL}$ (Baader et al., 2003; 2005) with additional concept and role constructors. These constructors do not have description logic semantics; instead, their meaning is completely determined by the set of generation rules of the CFQ dataset.

Let $A$ be a concept name, $C, C_1, C_2$ be concepts, $R, R_1, R_2$ be roles, and $v$ be a raw string. Then the following would be concepts:

```
C ::=      ⊤
      |    A
      |    C₁ ⊓ C₂
      |    ∃R.C
      |    And(C₁, C₂)
      |    DropDependency(C)
      |    Entity(v)
      |    PredicateWithBoundRolePairs(R₁, R₂)
      |    ProjectedRole(C₁, C₂)
      |    TypeInstance(C, v)
```

and the following would be roles:

```
R ::=      RolePair(C₁, C₂)
```

Note that our logical form does not have roles other than those in a form of `RolePair(C₁, C₂)`.

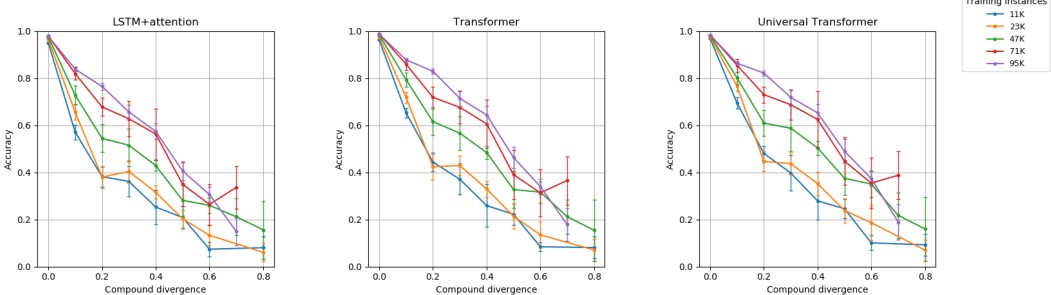

Figure 8: Accuracies of the three baseline systems on CFQ as a function of compound divergence at different training sizes.

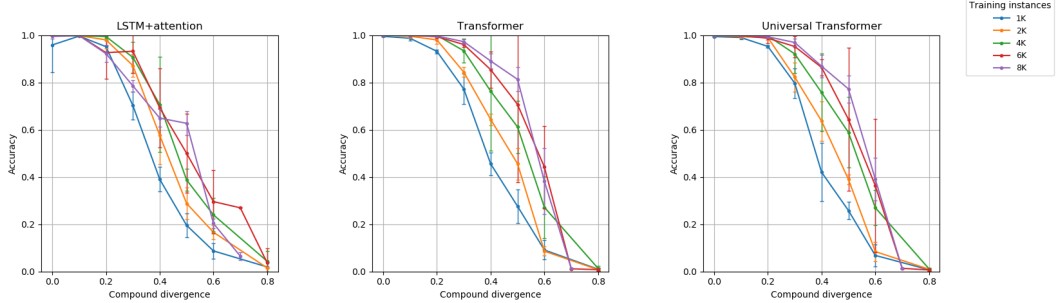

Figure 9: Accuracies of the three baseline systems on SCAN as a function of compound divergence at different training sizes.

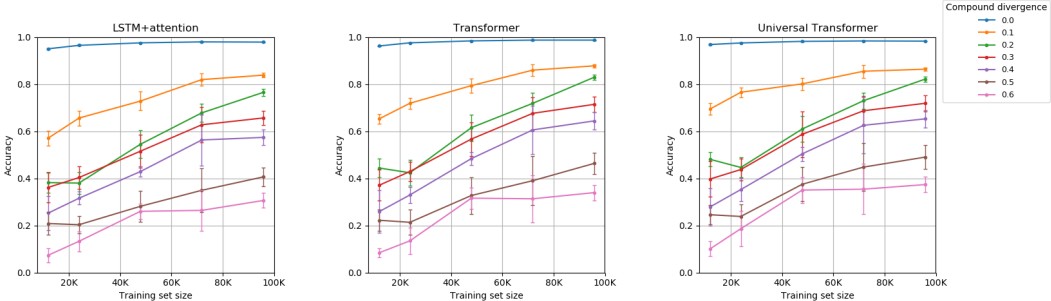

Figure 10: Accuracies of the three baseline systems on CFQ at different divergence levels as a function of training size.

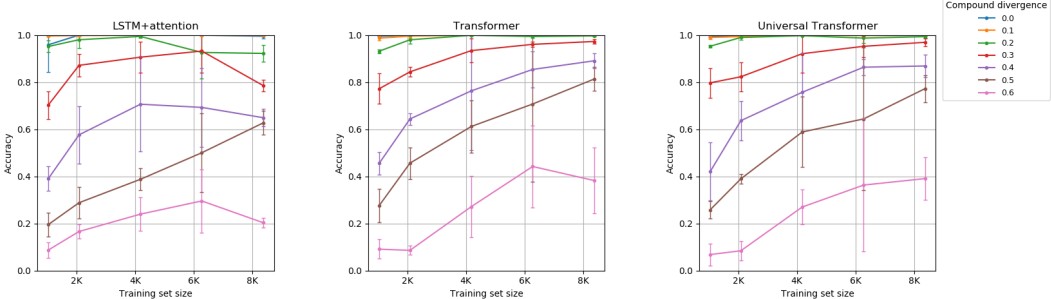

Figure 11: Accuracies of the three baseline systems on SCAN at different divergence levels as a function of training size.

New strings are generated by using a special function `new_var($S)`. This function generates a unique string of the form `?x<N>`, where N is a unique number, and assigns that string to variable $S. This string can later be used as a variable in a SPARQL constraint.

## J    RULE FORMAT

This section describes the format of each of the rule types we use for generating the CFQ dataset, in the form in which they appear in the rules index in Appendix M.

General formatting conventions shared across all rule types:

- Variable names are prefixed by '$'. Example: `$X`.
  (Exception: In grammar rules, while variables standing for constants are prefixed by '$', variables standing for logical forms are prefixed by '_'. Example: `_action`.)
- Concept names are written in camel case. Example: `FilmProducer`.
- Names of functions that output logical forms (concepts, roles, or knowledge) are also written in camel case. Examples: `DropDependency`, `BoundRolePairs`, `RolePair`.
- Names of functions that output string literals or which are used for converting logical forms to SPARQL are written in lowercase with underscores. Examples: `def2sparql`, `get_specializations`, `new_var`.
- String literals are enclosed in single quotes. Example: `'ns:film:director'`.

### J.1    GRAMMAR RULE FORMAT

The CFQ grammar is a unification-based grammar of recursive rewriting rules used to generate pairs of strings and their corresponding logical form. For an introductory overview of unification-based grammars including several popular variations, see Shieber (2003). The rules in the CFQ grammar follow a similar syntax in particular to that used in the Prolog extension GULP 3.1 (Covington, 1994), with the addition of support for disjunction, negation, absence, and default inheritance of features, and with minor differences in formatting described below.

Properties shared between the CFQ grammar syntax and that of (Covington, 1994) include the following:

- Grammar rules are notated as variations of context-free phrase-structure rules of the form $T_0 \rightarrow T_1 \; ... \; T_n$, where each of the syntactic non-terminals and terminals $T_0 \; ... \; T_n$ are augmented with feature lists in parentheses.
- Each grammar rule can be interpreted as specifying how a feature structure (with logical form) that is unifiable with the lefthand side can be re-written to the sequence of features structures (with logical form) indicated on the righthand side.
- Features are represented as attribute-value pairs separated by a colon (i.e., $attribute{:}value$).
- Shared values in feature structures are represented through the use of variables.

Specifically, in the rules index, CFQ grammar rules are described in the format

$$T_0(F_0)[H]/L_0 \rightarrow T_1(F_1)/L_1 \; ... \; T_n(F_n)/L_n$$

where:

- Each $T_i$ is a syntactic category (syntactic nonterminal) or a string literal (syntactic terminal).
- Each $L_i$ for $i \in [1, n]$ is either a variable representing a logical form or an empty string. In the case when $L_i$ is an empty string, we allow dropping the trailing slash from the $T_i(F_i)/L_i$ expression, resulting in just $T_i(F_i)$.
- $L_0$ is a logical form expressed in terms of $L_1...L_n$.

- Each $F_i$ is a comma-separated feature list of the form $(attribute_1:value_1, ..., attribute_k:value_k)$. In the case where $F_i$ is empty, we allow dropping the parentheses from the $T_i(F_i)$ expression, resulting in just $T_i$.

- $H$ is either an empty string or one of the variables $L_i$ for $i \in [1, n]$, indicating that $F_0$ default inherits the features of $F_i$ (the syntactic "head"). In the case where $H$ is an empty string, we allow dropping the brackets from the $T_0(F_0)[H]$ expression, resulting in just $T_0(F_0)$.

Note that while the above notation adopts the convention of splitting out the syntactic category and logical form from the feature list for visual prominence and to highlight the relationship to its context-free phrase-structure rule core, behaviorally it is identical to adding two more features to the feature list (we can call them, for example, $cat$ and $sem$) to represent the syntactic category and logical form.

This means that, for example, the rule

```
ACTIVE_VP[_head]/_head
→ VP_SIMPLE(form:infinitive)/_head
```

can be considered a notational shorthand for the following rule expressed purely using feature lists:

```
(cat:ACTIVE_VP, sem:_head)[_head]
→ (cat:VP_SIMPLE, sem:_head, form:infinitive)
```

**Disjunction of features.** Similarly to (Karttunen, 1984), we allow disjunctive feature specifications, which we denote by separating the alternative values with a pipe ('|'). The feature specification (form:gerund|infinitive) would thus unify with either (form:gerund) or (form:infinitive), but not with (form:past_participle).

**Absence of features.** We use a special atomic value _none_ to indicate that a given feature must either be absent or else explicitly set to the value _none_. The feature specification (subject:_none_, object:yes) would thus unify with either (object:yes) or (subject:_none_, object:yes), but not with (subject:yes, object:yes).

**Negation of features.** Similarly to (Karttunen, 1984), we allow negated feature specifications, which we denote by prefixing the attribute with a minus sign ('-'). The feature specification (-form:gerund|infinitive) would thus unify with (form:past_participle) or (form:_none_), but not with (form:gerund) or (form:infinitive). In general, a feature specification of the form (-attribute:$v_1$|...|$v_j$) can be considered a notational shorthand for (attribute:$v_{j+1}$|...|$v_k$|_none_), where $v_{j+1}$|...|$v_k$ is an enumeration of all possible values of the feature attribute other than $v_1$|...|$v_j$.

**Default inheritance of features.** If the lefthand side term is notated as $T_0(F_0)[H]$, with $H$ equal to one of the variables $L_i$ for $i \in [1, n]$, then this is interpreted as a notational shorthand for augmenting both $F_0$ and $F_i$ with an additional list of attribute-value pairs $(a_1:\$v_1, ..., a_k:\$v_k)$, where $a_1...a_k$ are all of the attributes listed in $F_i$ that were not originally listed in $F_0$.

**Unification of logical forms.** As described in Appendix I, we represent logical forms using a variation of description logic, rather than using feature structures. In the context of unification, we consider logical forms to unify if and only they achieve structural concept equality after variable replacement (using the same variable replacements applied during unification of the corresponding feature lists), while taking into account the commutativity and associativity of ⊓. For example, under this criterion, the logical form GenderRel ⊓ ∃RolePair(Predicate, Gender)._head would unify with either GenderRel ⊓ ∃RolePair(Predicate, Gender).Male or with (∃RolePair(Predicate, Gender).Male) ⊓ GenderRel under a variable replacement mapping _head to Male, but would not unify with GenderRel ⊓ ∃RolePair(Predicate, Gender).Male ⊓ ∃RolePair(Predicate, GenderHaver).FilmProducer.

## J.2 KNOWLEDGE RULE FORMAT

CFQ knowledge rules output expressions representing facts that are known to be true. They have no direct effect on text, logical forms, or SPARQL, but the generated knowledge can be used as preconditions to other rules. In the rules index, they are described in the following format:

$\rightarrow K$, where $K$ is knowledge that is output.

By convention, we define the rule name of a knowledge rule to be simply the string representing the knowledge that the rule outputs, and we omit the rule name in the rules index for brevity.

The union of those rules defines a knowledge base which we denote with $KB^{CFQ}$.

All knowledge in CFQ is represented in the form $P(X_1, ..., X_n)$, where $P$ is a predicate from the list below, and $X_1, ..., X_n$ are either logical forms or else raw strings. Knowledge rules do not use variable-based expressions.

Supported knowledge predicates:

- BoundRolePairs
- ExclusiveRolePair
- FreebaseEntityMapping
- FreebasePropertyMapping
- FreebaseTypeMapping
- NonExclusiveRolePair
- Role

## J.3 INFERENCE RULE FORMAT

CFQ inference rules transform logical forms and may be conditioned on knowledge. In the rules index, they are described in the following format:

$K : L_0 \rightarrow L_1$

where $K$ represents a comma-separated list of knowledge preconditions, and $L_0$ and $L_1$ represent the input and output logical forms, all expressed in terms of a shared set of variables $v_1, ..., v_m$.

These rules are interpreted as stating that if there exists a variable replacement $r()$ replacing $v_1, ..., v_m$ with some logical forms $l_1, ..., l_m$ respectively, such that $r(K) \subseteq KB^{CFQ}$, then we can apply the inference rule by rewriting $r(L_0)$ to $r(L_1)$.

## J.4 RESOLUTION RULE FORMAT

CFQ resolution rules transform SPARQL expressions and may be conditioned on knowledge. They do not affect text or logical forms.

In the rules index, they are described in the following format:

$K : S_0 \rightarrow S_1 ... S_n$

where $K$ represents a comma-separated list of knowledge preconditions, $S_0$ is a variable-based expression and $S_1 ... S_n$ are either raw SPARQL strings or else expressions described in terms of the same variables used in $S_0$ and $K$.

These rules are interpreted as stating that if there exists a variable replacement $r()$ replacing $v_1, ..., v_m$ with some logical forms, strings, or expressions $l_1, ..., l_m$ respectively, such that $r(K) \subseteq KB^{CFQ}$, then we can apply the resolution rule by rewriting $r(S_0)$ to the sequence of terms $r(S_1) ... r(S_n)$.

## K  GENERATION ALGORITHM

Our generation algorithm produces triples of the form ⟨question, logical form, SPARQL query⟩ in a mixed top-down and bottom-up fashion, with the final program of rule applications output alongside each triple in the form of a rule application DAG. The top-down portion of generation is responsible for efficiently searching for rules that can be applied to produce a meaningful example, while the bottom-up portion is responsible for actually applying the rules (i.e., performing the composition) and for producing the DAG.

The generation process proceeds in two phases, each involving a top-down as well as bottom-up aspect. In the first phase, we apply grammar rules interleaved with inference rules to produce a pair of ⟨question, logical form⟩. Specifically, we apply a recursive top-down algorithm which starts with the $S$ nonterminal and at every step performs a random search over the rules in the grammar which could produce the target nonterminal with accompanying feature structure. This top-down process proceeds until a candidate syntactic parse tree is attained whose leaves consist purely of syntactic terminals (i.e., string literals or entity placeholders). The grammar rules from this candidate parse tree are then applied in a bottom-up fashion beginning with the syntactic terminals to yield a tree of ⟨text, logical form⟩ pairs. After each such bottom-up grammar rule application, we then greedily apply all possible inference rules on the resulting logical forms, applying an arbitrary deterministic ordering to the inference rules in cases where rules could be applied in multiple valid orderings. This ensures that inference rules and grammar rules are executed in an interleaved manner and each inference rule is applied at the earliest possible occasion.

When a ⟨question, logical form⟩ pair is generated for the $S$ nonterminal, we proceed to the second phase of the algorithm, in which resolution rules are applied to generate a corresponding SPARQL query to make up the third element of the desired ⟨question, logical form, SPARQL query⟩ triple. In practice, the bulk of the work in this phase is performed in a top-down fashion, in which resolution rules are recursively applied to transform a starting expression of the form `get_specializations($L)` (where `$L` represents the logical form output from the grammar phase) into a sequence of text literals representing the SPARQL query. This is followed nominally by a bottom-up process to construct the rule application DAG, yielding a tree of resolution rule applications of a similar form to the tree of interleaved grammar and inference rules output from the grammar phase. Note that while the grammar phase involves a large degree of random choice, the resolution phase proceeds much more deterministically, as the CFQ resolution rules have been designed such that any given question can yield only one possible SPARQL query, modulo commutativity and associativity of ⊓. In cases where resolution rules could be applied in multiple valid orderings, we again apply an arbitrary deterministic ordering to the resolution rules so as to yield as consistent as possible a rule application DAG and ⟨question, logical form, SPARQL query⟩ triple for any given question.

Finally, to ease the task of tracking unique query patterns and to minimize the impact on the learning task of implementation details regarding choice of variable names or ordering of clauses, we normalize the final SPARQL query by alphabetically sorting the query clauses and re-numbering the variables to follow a standard increasing order.

The resulting ⟨question, logical form, SPARQL query⟩ triple is then appended to the CFQ dataset.

### K.1  JOIN BY LOGICAL FORM

In general, we do not explicitly track rules to represent the example-independent behaviors of the generation algorithm, as the universal applicability of these rules mean that the complete behavior of the generator should be observable on any reasonably-sized train set. The same applies to certain core behaviors of the description logic $\mathcal{EL}$, such as commutativity and associativity of ⊓, which we omit tracking as explicit rules due to their similar ubiquity of application.

One example-independent rule, however, that we do explicitly track is the rule that describes the handover process between the grammar phase and the resolution phase – or in terms of the rule application DAG, the rule that joins the tree of interleaved grammar and inference rule applications with the tree of resolution rule applications. We call this rule `JOIN_BY_LOGICAL_FORM`. It is included in the rules list for every example in CFQ and appears as the head of the rule application tree for each example.

## K.2   RELATIONSHIP BETWEEN GENERATION AND PARSING

Note that conceptually a similar approach for combining the different rule types could be applied to the semantic parsing task. The main difference would be that, instead of performing random search over the grammar, the semantic parsing task would need to find the set of rules which produce the desired input text.

## K.3   SELECTING AN APPROPRIATE SAMPLE SET

For many domains, the set of examples generated by exhaustively combining rules is infinite or prohibitively large. For example, the CFQ grammar generates an infinite set of questions, and even when restricted to a reasonable complexity, the set is still too large for practical use. This means that we need to choose which subset of examples we want to include in our dataset. Given our goal of comprehensively measuring compositional generalization, we do this by:

1. maximizing the overall diversity of rule combinations (allowing us to test as many rule combinations as possible)

2. while using a uniform distribution from simple examples to increasingly more complex examples.

We measure the diversity of rule combinations of a dataset using the empirical entropy over the frequency distribution of the subgraphs of the rule application DAGs, and we measure the complexity of an example using the number of rule applications used to generate it.

For CFQ, we choose the following practical trade-off between these two criteria. We first generate a sufficiently large sample set by performing random rule applications. We then subsample from it to select a subset that maximizes the entropy of the subgraph distribution (while only taking into account subgraphs with a limited number of nodes for practicality). We use a greedy algorithm that incrementally assigns elements to the subsampled set while maximizing entropy at each step.

The subsampling is initially limited to examples with the smallest complexity level and continues with increasingly larger complexity levels. We cap the maximum number of examples per level to achieve a uniform distribution across levels, and we limit the maximum complexity level such that the questions remain relatively natural. Table 1 shows examples of generated questions at varying levels of complexity.

## L   EXAMPLE OF A RULE APPLICATION DAG

Figures 12 through 14 show the rule application DAG that was produced when generating the question "Who directed [entity]?". They illustrate how grammar, inference, and knowledge rules are combined to generate a pair of text and logical form, and how resolution rules are used to generate the SPARQL query for the resulting logical form.

## L.1   DAG NORMALIZATION

As discussed in Section 3, nodes of this DAG represent rule applications while edges represent dependencies among the rules; i.e., an edge $A \rightarrow B$ means that rule $B$ strictly depends on rule $A$ in the sense that the generator cannot apply rule $B$ before applying rule $A$. The DAG is normalized to ensure that a certain rule combination is represented using the same DAG across all the examples where it occurs. This is important for meaningfully comparing measures such as entropy and divergence across subgraphs of different examples.

Specifically, together with adopting the measures described above to ensure that rules are applied in a deterministic order, we achieve the normalization of the DAG by only producing edges that represent "minimal dependencies". This means that if a rule $A$ can be applied after rule $B$, but it could also be applied after rule $B'$ with $B \rightarrow B'$ (i.e., $B'$ depends on $B$), we don't produce the edge $B' \rightarrow A$.

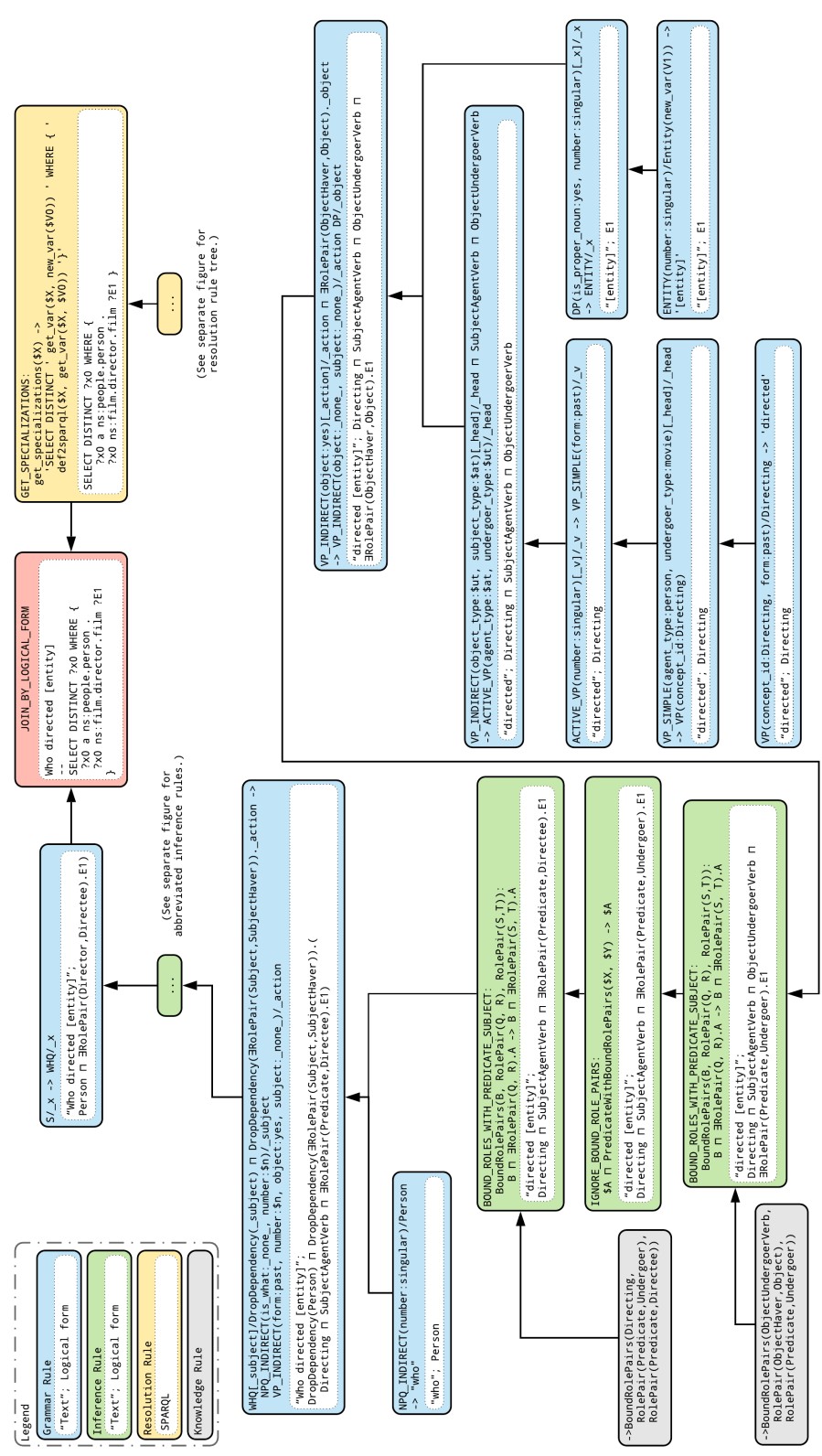

Figure 12: The normalized rule application DAG that was produced for "Who directed [entity]?" (grammar/inference rules portion, continued in Figures 13 and 14).

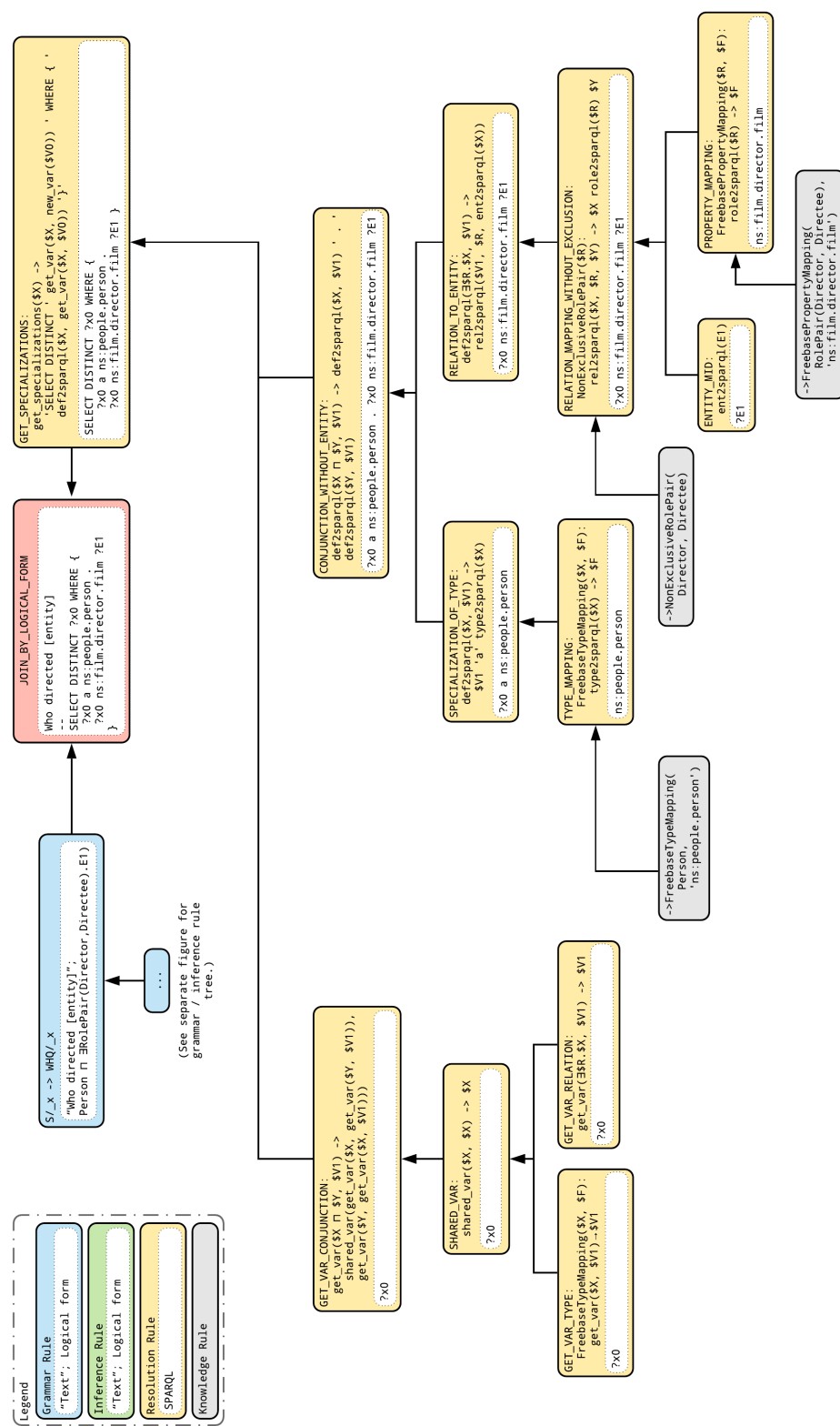

Figure 13: The normalized rule application DAG that was produced for "Who directed [entity]?" (resolution rules portion, continued from Figure 12).

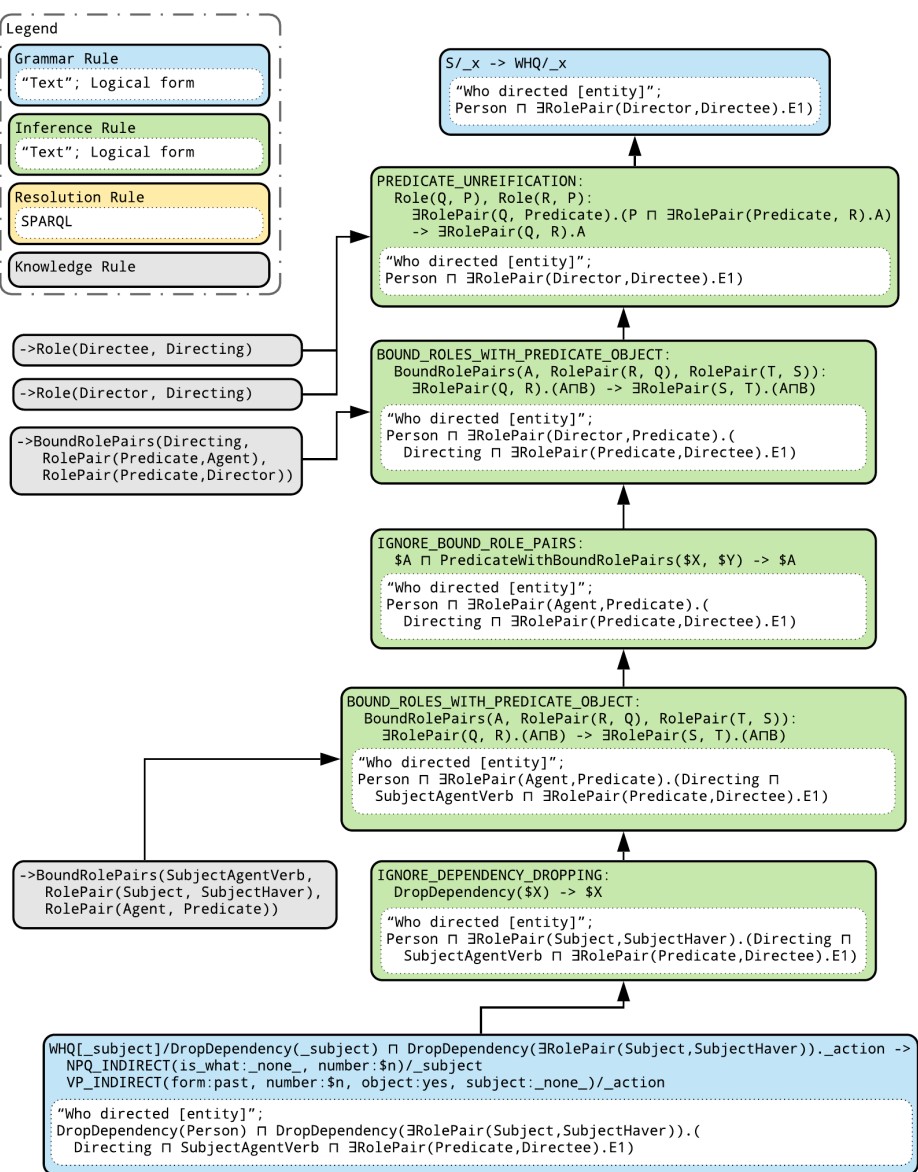

Figure 14: The normalized rule application DAG that was produced for "Who directed [entity]?" (inference rules portion, continued from Figure 12).

## L.2 CONCEPT ABBREVIATIONS

For brevity, in the rule application DAG figures we have applied the following abbreviations for several lengthy concept names:

- `Director = FilmDirector`

- `Directee = DirectedFilm`

- `Directing = DirectingAFilm`

- `SubjectAgentVerb   =   PredicateWithBoundRolePairs(RolePair(SubjectHaver, Subject), RolePair(Predicate, Agent))`

- `ObjectUndergoerVerb = PredicateWithBoundRolePairs(RolePair(ObjectHaver, Object), RolePair(Predicate, Undergoer))`
- `E1 = Entity('?E1')`

### L.3 ENTITY PLACEHOLDERS

As described in Section 3.2, during generation we initially generate a ⟨question, logical form, SPARQL query⟩ triple containing entity placeholders, and then replace those placeholders with specific entities as a post-processing step. Conceptually, one could construct a rule application DAG describing either the process by which the original ⟨question, logical form, SPARQL query⟩ triple with entity placeholders was generated, or alternatively the rules that would need to be applied if constructing the ⟨question, logical form, SPARQL query⟩ triple containing the final entity MIDs directly. Structurally, these two DAGs are identical, differing only in the definition of two entity-related rules described below. The rule application DAG shown in the accompanying figures is the version using entity placeholders.

**Versions of entity rules applicable when using entity placeholders:**

**ENTITY=[ENTITY]_HSz7QrdGdsX**:
```
ENTITY(number:singular)/Entity(new_var(V1))
→ '[entity]'
```
**ENTITY_MID**:
```
ent2sparql(Entity($X)) → $X
```

**Versions of entity rules applicable when using actual entity MIDs:**

**ENTITY=[ENTITY]_HSz7QrdGdsX**:
```
ENTITY(number:singular)/'m.'$X
→ 'm.'$X
```
**ENTITY_MID**:
```
ent2sparql('m.'$X) → 'ns:m.'$X
```

### L.4 SUBGRAPHS AND THEIR WEIGHTS

Figure 15 shows an example of subgraphs in order to provide more details on the sampling and weighting of compounds. An example non-linear subgraph is highlighted by the red area, and two linear subgraphs are highlighted by the blue and the yellow areas, respectively.

As described in Section 2.1, given a large subset $\mathbb{G}$ of subgraphs from the sample set as a whole, we calculate for each sample the weight of each subgraph $G \in \mathbb{G}$ that occurs in that sample as:

$$w(G) = \max_{g \in \mathrm{occ}(G)} \big(1 - \max_{G':g \prec g' \in \mathrm{occ}(G')} P(G'|G)\big),$$

where $\mathrm{occ}(G)$ is the set of all occurrences of $G$ in the sample, $\prec$ denotes the strict subgraph relation, and $P(G'|G)$ is the empirical probability of $G'$ occurring as a supergraph of $G$ over the full sample set.

Intuitively, we are trying to estimate how interesting the subgraph $G$ is in the sample. First, for every occurrence $g$ of a subgraph $G$, we look for the supergraph $G'$ of $g$ that co-occurs most often with $G$ in the full sample set. The empirical probability of having $G'$ as a supergraph of $G$ determines how interesting the occurrence $g$ is – the higher this probability, the less interesting the occurrence. Thus we compute the weight of the occurrence as the complement of this maximum empirical probability. Then we take the weight of $G$ to be the weight of the most interesting occurrence $g$ of $G$ in the sample.

E.g. in the extreme case that $G$ *only* occurs within the context $G'$, the weight of $G$ will be 0 in all samples. Conversely, if $G$ occurs in many different contexts, such that there is no single other subgraph $G'$ that subsumes it in many cases, then $w(G)$ will be high in all samples in which it occurs. This ensures that when calculating compound divergence based on a weighted subset of compounds, the

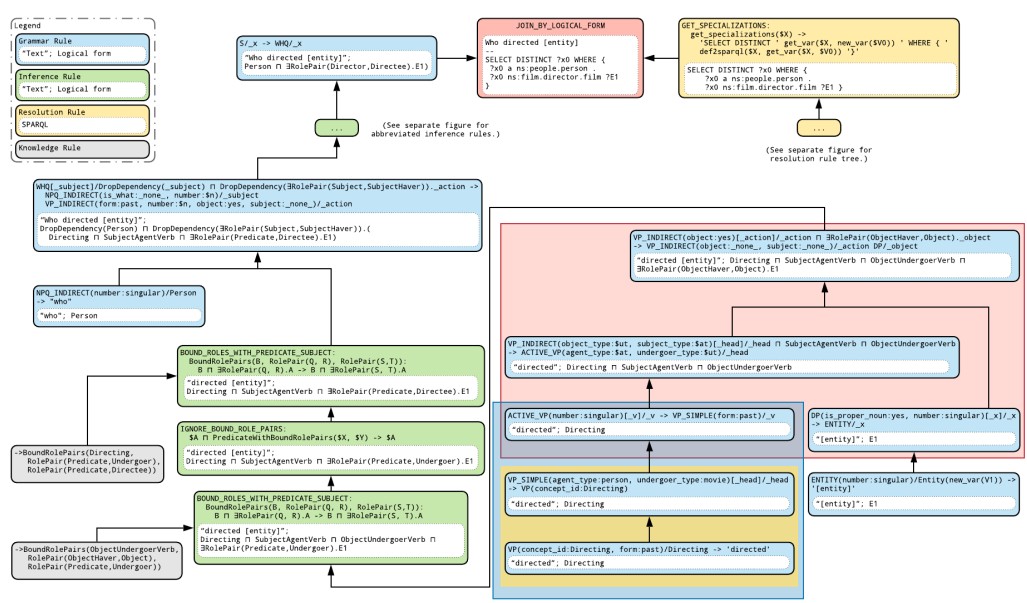

Figure 15: Examples subgraphs in the grammar/inference rules portion for "Who directed [entity]?" (from Figure 12): non-linear subgraph (red area), and two linear subgraphs (yellow and blue areas), of which one (yellow area) is a subgraph of the other (blue area).

most representative compounds are taken into account, while avoiding double-counting compounds whose frequency of occurrence is already largely explainable by the frequency of occurrence of one of its super-compounds.

Returning to our example in Figure 15, suppose that $G$ represents the smallest linear subgraph (yellow area), and suppose that the weight of $G$ in this sample is 0.4. Then this means that there exists some other subgraph $G'$ (for instance, the linear subgraph highlighted by the blue area) that is a supergraph of $G$ in 60% of the occurrences of $G$ across the sample set.

## M   RULES INDEX

Below is a selection of the rules used in the generation of CFQ. Specifically, this includes all rules involved in generating the question "Who directed [entity]?" (the same example illustrated in the rule application DAG in Appendix L). The format of the rules is discussed in Appendix J.

### M.1   GRAMMAR RULES

**S=WHQ_F6E9egkQqxj**:
```
 S/_x
→ WHQ/_x
```

**WHQ=NPQ_INDIRECT_VP_INDIRECT_TXCca9URgVm**:
```
 WHQ[_subject]/DropDependency(_subject) ⊓
DropDependency(∃RolePair(Subject, SubjectHaver)._action)
→ NPQ_INDIRECT(is_what:_none_, number:$n)/_subject
   VP_INDIRECT(form:past, number:$n, object:yes, subject:_none_)/_action
```

**NPQ_INDIRECT=WHO_5ptbPXXbuLZ**:
```
 NPQ_INDIRECT(number:singular)/Person
→ 'who'
```

**VP_INDIRECT=VP_INDIRECT_DP_ZJH4NhRkByc**:

```
 VP_INDIRECT(object:yes)[_action]/_action ⊓ ∃RolePair(ObjectHaver,
Object)._object
  → VP_INDIRECT(object:_none_, subject:_none_)/_action
    DP/_object
```

**VP_INDIRECT=ACTIVE_VP_RX51Tm7RXPe**:
```
 VP_INDIRECT(object_type:$ut, subject_type:$at)[_head]/_head ⊓
PredicateWithBoundRolePairs(RolePair(SubjectHaver, Subject),
RolePair(Predicate, Agent)) ⊓
PredicateWithBoundRolePairs(RolePair(ObjectHaver, Object),
RolePair(Predicate, Undergoer))
  → ACTIVE_VP(agent_type:$at, undergoer_type:$ut)/_head
```

**ACTIVE_VP=VP_SIMPLE_hJqAyjRUYJp**:
```
 ACTIVE_VP(number:singular)[_head]/_head
  → VP_SIMPLE(form:past)/_head
```

**VP_SIMPLE=VP_GHWf3fcVRZg**:
```
 VP_SIMPLE(agent_type:person, undergoer_type:movie)[_head]/_head
  → VP(concept_id:DirectingAFilm)/_head
```

**VP=DIRECTED_JkYzNbQyXtv**:
```
 VP(concept_id:DirectingAFilm, form:past)/DirectingAFilm
  → 'directed'
```

**DP=ENTITY_M6fSP5GvRaN**:
```
 DP(is_proper_noun:yes, number:singular)[_head]/_head
  → ENTITY/_head
```

**ENTITY=[ENTITY]_HSz7QrdGdsX**:
```
 ENTITY(number:singular)/Entity(new_var(V1))
  → '[entity]'
```

... (211 grammar rules total)

## M.2   INFERENCE RULES

**BOUND_ROLES_WITH_PREDICATE_OBJECT**:
```
 BoundRolePairs($A, RolePair($R, $Q), RolePair($T, $S)):
∃RolePair($Q, $R).($A ⊓ $B) → ∃RolePair($S, $T).($A ⊓ $B)
```

**BOUND_ROLES_WITH_PREDICATE_SUBJECT**:
```
 BoundRolePairs($B, RolePair($Q, $R), RolePair($S, $T)):
$B ⊓ ∃RolePair($Q, $R).$A → $B ⊓ ∃RolePair($S, $T).$A
```

**IGNORE_BOUND_ROLE_PAIRS**:
```
 $A ⊓ PredicateWithBoundRolePairs($X, $Y) → $A
```

**IGNORE_DEPENDENCY_DROPPING**:
```
 DropDependency($X) → $X
```

**PREDICATE_UNREIFICATION**:
```
 Role($Q, $P), Role($R, $P):
∃RolePair($Q, Predicate).($P ⊓ ∃RolePair(Predicate, $R).$A) →
∃RolePair($Q, $R).$A
```

... (17 inference rules total)

## M.3   RESOLUTION RULES

**CONJUNCTION_WITHOUT_ENTITY**:
```
 def2sparql($X ⊓ $Y, $V1) → def2sparql($X, $V1) ' . ' def2sparql($Y,
$V1)
```

**ENTITY_MID**:
```
ent2sparql(Entity($X)) → $X
```

**GET_SPECIALIZATIONS**:
```
get_specializations($X) → 'SELECT DISTINCT ' get_var($X, new_var($V0))
' WHERE { ' def2sparql($X, get_var($X, $V0)) '}'
```

**GET_VAR_CONJUNCTION**:
```
get_var($X ⊓ $Y, $V1) → shared_var(get_var($X, get_var($Y, $V1)),
get_var($Y, get_var($X, $V1)))
```

**GET_VAR_RELATION**:
```
get_var(∃$R.$X, $V1) → $V1
```

**GET_VAR_TYPE**:
```
FreebaseTypeMapping($X, $F):
get_var($X, $V1) → $V1
```

**PROPERTY_MAPPING**:
```
FreebasePropertyMapping($R, $F):
role2sparql($R) → $F
```

**RELATION_MAPPING_WITHOUT_EXCLUSION**:
```
NonExclusiveRolePair($R):
rel2sparql($X, $R, $Y) → $X role2sparql($R) $Y
```

**RELATION_TO_ENTITY**:
```
def2sparql(∃$R.$X, $V1) → rel2sparql($V1, $R, ent2sparql($X))
```

**SHARED_VAR**:
```
shared_var($X, $X) → $X
```

**SPECIALIZATION_OF_TYPE**:
```
def2sparql($X, $V1) → $V1 ' a ' type2sparql($X)
```

**TYPE_MAPPING**:
```
FreebaseTypeMapping($X, $F):
type2sparql($X) → $F
```

... (21 resolution rules total)

## M.4 KNOWLEDGE RULES

```
 → BoundRolePairs(DirectingFilm, RolePair(Predicate, Agent),
RolePair(Predicate, FilmDirector))
 → BoundRolePairs(DirectingFilm, RolePair(Predicate, Undergoer),
RolePair(Predicate, DirectedFilm))
 → BoundRolePairs(PredicateWithBoundRolePairs(RolePair(ObjectHaver,
Object), RolePair(Predicate, Undergoer)), RolePair(ObjectHaver, Object),
RolePair(Predicate, Undergoer))
 → BoundRolePairs(PredicateWithBoundRolePairs(RolePair(Subject,
SubjectHaver), RolePair(Agent, Predicate)), RolePair(Subject,
SubjectHaver), RolePair(Agent, Predicate))
 → FreebasePropertyMapping(RolePair(FilmDirector, DirectedFilm),
'ns:film.director.film')
 → FreebaseTypeMapping(Person, 'ns:people.person')
 → NonExclusiveRolePair(FilmDirector, DirectedFilm)
 → Role(DirectedFilm, DirectingFilm)
 → Role(FilmDirector, DirectingFilm)
```

... (194 knowledge rules total)

