# OpenReview forum: "Measuring Compositional Generalization: A Comprehensive Method on Realistic Data"
_ICLR.cc/2020/Conference — Accept (Poster)_

### Official Review · AnonReviewer2 · 2019-10-15
**Official Blind Review #2**

**Rating:** 6

**Review:**

Summary

Strength:
1. This topic studied in this paper is interesting and is helping to promote the following developing of algorithms with compositional generalization ability.
2. The experimental results show the effectiveness of the proposed metric for measuring compositional diversity.


comments:
1. How to control the trade-off between the atom and compositional divergence? It's interesting to show how different
compositional divergence can affect the performance of different models.
2. Many previous works are proposed for improving the generalization ability of the seq2seq models[1]. More experiments need to
be conducted using these previous methods.

[1]Compositional generalization in a deep seq2seq model by separating syntax and semantics


**Experience Assessment:**

I do not know much about this area.

**Review Assessment: Checking Correctness Of Derivations And Theory:**

I assessed the sensibility of the derivations and theory.

**Review Assessment: Checking Correctness Of Experiments:**

I assessed the sensibility of the experiments.

**Review Assessment: Thoroughness In Paper Reading:**

I read the paper at least twice and used my best judgement in assessing the paper.

---

> ### Author Response · Authors · 2019-11-13
> **Response to reviewer**
>
> Thank you for your review and for highlighting the strengths of the paper.
>
> Regarding comment (1), as described in Section 4, splitting is performed using an iterative greedy algorithm that alternately adds examples to the training and test sets or removes examples from them, while selecting at each step the element that brings the atom and compound divergences as close as possible to their desired values. As we always target a low atom divergence (specifically <= 0.02, as described in Sections 4 and 5.1), the atom divergence is treated in effect as a constraint, within the bounds of which the splitting algorithm seeks to optimize the compound divergence to match the target divergence as closely as possible. This formulation makes the balancing of the two criteria during the course of splitting relatively straightforward: we penalize deviation from the target atom divergence only in the case that the atom divergence exceeds its target threshold, and we penalize such deviation from the target atom divergence significantly more strictly than deviation from the target compound divergence, so as to minimize the chance that the target atom divergence is exceeded at completion of splitting.
>
> Regarding showing how different compositional divergence can affect the performance of different models, we illustrate this in Figure 2 and in Section 5, where we construct a series of divergence-based splits with different compound divergences and demonstrate that there is a strong negative correlation between the compound divergence and the performance of each of the three baseline models.
>
> If this is not clear, please let us know, and we would be happy to clarify further.
>
> Regarding comment (2), please note that the paper that you mention was already cited in our original submission. We agree with you that evaluating the Syntactic Attention Model on our dataset is an interesting direction of future work, which we now mention more clearly in the paper (see Section 7). However, it is not the focus of our paper to investigate any particular architecture or model that is optimized for its compositional generalization abilities, but instead we present a framework in which this ability can be measured comprehensively. With the release of our benchmark we hope that it will be used (by us, and by other researchers) to investigate which learning systems and approaches show most promise for compositional generalization.

---

### Official Review · AnonReviewer1 · 2019-10-24
**Official Blind Review #1**

**Rating:** 8

**Review:**

This paper first introduces a method for quantifying to what extent a dataset split exhibits compound (or, alternatively, atom) divergence, where in particular atoms refer to basic structures used by examples in the datasets, and compounds result from compositional rule application to these atoms. The paper then proposes to evaluate learners on datasets with maximal compound divergence (but minimal atom divergence) between the train and test portions, as a way of testing whether a model exhibits compositional generalization, and suggests a greedy algorithm for forming datasets with this property. In particular, the authors introduce a large automatically generated semantic parsing dataset, which allows for the construction of datasets with these train/test split divergence properties. Finally, the authors evaluate three sequence-to-sequence style semantic parsers on the constructed datasets, and they find that they all generalize very poorly on datasets with maximal compound divergence, and that furthermore the compound divergence appears to be anticorrelated with accuracy.

This is an interesting and ambitious paper tackling an important problem. It is worth noting that the claim that it is the compound divergence that controls the difficulty of generalization (rather than something else, like length) is a substantive one, and the authors do provide evidence of this. At the same time, I think the authors could possibly do more to show that the trend in the plots in Figure 2 can't be explained by something else: for example, the authors could show that the length ratios remain constant as the compound divergence is varied.  I think it is also not necessarily clear how easily the notion of differing compound distributions generalizes to other types of tasks.

Presentation-wise, much of the paper is clear and well written, though I think the discussion of weighted frequency distributions of compounds (top of page 3) could be clarified further, and in particular an example subgraph of a rule application DAG should be highlighted here.

**Experience Assessment:**

I have read many papers in this area.

**Review Assessment: Checking Correctness Of Derivations And Theory:**

I assessed the sensibility of the derivations and theory.

**Review Assessment: Checking Correctness Of Experiments:**

I assessed the sensibility of the experiments.

**Review Assessment: Thoroughness In Paper Reading:**

I read the paper at least twice and used my best judgement in assessing the paper.

---

> ### Author Response · Authors · 2019-11-13
> **Response to reviewer**
>
> Thank you for reading our paper in depth and for the interesting comments and observations.
>
> Addressing the specific comments in order:
>
> On (other) explanations for the trend between compound divergence and accuracy which we observe in Figure 2 ("show that the length ratios remain constant as the compound divergence is varied"): We did some further specific analysis to show that length variation (as measured by the length ratios) is not a better explanation for the drop in accuracy. We added a paragraph discussing this to the paper in Section 5.2 now. Also, we had already observed in Table 3 and the discussion that we did not expect the lengths to stay perfectly constant between train and test splits. Note, however, that the splitting algorithm could be extended in a way that would try to achieve maximum compound divergence while keeping both the atom divergence close to zero and the length distribution constant between the splits. Here, we focus on the compound divergence as a single measure of compositionality challenge. (Compare also the discussion in Section 4: "Interestingly, the MCD splits still correlate with the aspects of compositional generalization that are targeted by the other experiments in this table. As shown in the four right columns of Table 3, for each MCD split, the train set V contains on average shorter examples than the test set W (measured by the ratio of average lengths), and [...]. However, these correlations are less pronounced than for the experiments that specifically target these aspects, and they vary significantly across the different MCD splits.")
>
> On using the idea of varying compound divergence for other tasks ("not necessarily clear how easily the notion of differing compound distributions generalizes to other types of tasks"): We agree that this is an interesting direction for future work. For example, we believe that a similar approach may be applicable to visual tasks for which the compounds can be programmatically analyzed, e.g. visual question answering tasks like CLEVR. We mention this now explicitly in Section 7. The same way that splits based on lengths or patterns have been used for natural language compositionality analysis, vision tasks have used specific splits for analysis of compositionality e.g. based on color or pairs of objects in images. (We discuss this in Section 6 to some degree.) Therefore it seems a reasonable expectation to us that the DBCA approach would also transfer to such tasks.
>
> Regarding the "discussion of weighted frequency distributions of compounds": Thank you for your suggestion to clarify this section.
>
> We have reworded the relevant paragraph to make it more precise. We have also added illustrative examples of subgraphs in Appendix L.4, together with a more detailed explanation of the weight calculation. We hope that the more precise presentation and the discussion in the appendix help to clarify this point.

---

### Official Review · AnonReviewer3 · 2019-10-24
**Official Blind Review #3**

**Rating:** 6

**Review:**

This paper introduces a method for generating training/test data for measuring the model's ability of "compositional generalization" in complex compositional tasks/domains such as natural language understanding, visual understanding and other domains.
The idea of the proposed method is to essentially keep the "atom" distribution unchanged between the training and test data, but maximize the divergence between the "compound" distributions between them.
The authors have conducted a thorough and systematic experimentation, comparing the proposed approach with a number of other heuristic approaches for train test splits (such as random and input/output length, etc.) and using both a new large data set they generated (CFQ) and existing data set (SCAN).
The experimental results verify that using their method they can obtain train test data sets with uniform atom distributions with large divergence in compound distributions, and they find that there is a surprisingly large negative correlation between the accuracy of existing state-of-the-art learning methods and the compound divergence.
The data generation mechanism is systematic and involved, consisting of different categories of rules (logical form, grammar, rule application DAG's, etc.) and it would seem that the generation method/system and the generated data would be useful as benchmark data for the community.
The paper lacks technical novelty other than the training and test data generation approach, but having one available to the community with these apparently desirable characteristics as benchmark data for measuring complex, compositional generalization capabilities, and that could be invaluable to the research community.

**Experience Assessment:**

I have published one or two papers in this area.

**Review Assessment: Checking Correctness Of Derivations And Theory:**

N/A

**Review Assessment: Checking Correctness Of Experiments:**

I assessed the sensibility of the experiments.

**Review Assessment: Thoroughness In Paper Reading:**

I read the paper at least twice and used my best judgement in assessing the paper.

---

> ### Author Response · Authors · 2019-11-13
> **Response to reviewer**
>
> We thank the reviewer for reading our paper in detail and for providing a careful review of the paper along with a summary of the contributions. We agree with the reviewer that some of the primary areas of technical novelty in the paper are related to the training and test data generation approach. More specifically, we would suggest that the novelty consists of at least two aspects: one being the approach to generation of a structurally diverse dataset through maximization of the entropy of the compound distribution; the second (and in our opinion more fundamental) novel aspect being the DBCA approach to measuring compositional generalization capability through the creation of data splits of varied compound divergence (while keeping atom distribution similar).

---

### Author Response · Authors · 2019-11-13
**Response to all reviewers**

We thank the reviewers for the careful reading of the paper and the valuable comments. We have revised the paper as suggested by the reviewers, and summarize the major changes as follows:

* Reworded the paragraph in Section 2.1 discussing subgraph weighting to make it more precise, and added an Appendix L.4 with illustrative examples of subgraphs, together with a more detailed explanation of the weight calculation.
* Added a paragraph in Section 5.2 with further analysis to show that length variation (as measured by the length ratios) is not a better explanation for the drop in accuracy.
* Expanded the discussion of future work slightly.

In addition we made the following smaller changes:
* Corrected a few typos.
* Small layout and text changes to keep the paper within the 10-page limit.
* Updated the example dataset item in Appendix A to better match the format that we plan to use for the data release.
* Corrected some details of the qualitative error analysis in Appendix F.2.

---

### Decision · Program_Chairs · 2019-12-19

**Decision:**

Accept (Poster)

**Comment:**

Main content:

Blind review #1 summarizes it well:

This paper first introduces a method for quantifying to what extent a dataset split exhibits compound (or, alternatively, atom) divergence, where in particular atoms refer to basic structures used by examples in the datasets, and compounds result from compositional rule application to these atoms. The paper then proposes to evaluate learners on datasets with maximal compound divergence (but minimal atom divergence) between the train and test portions, as a way of testing whether a model exhibits compositional generalization, and suggests a greedy algorithm for forming datasets with this property. In particular, the authors introduce a large automatically generated semantic parsing dataset, which allows for the construction of datasets with these train/test split divergence properties. Finally, the authors evaluate three sequence-to-sequence style semantic parsers on the constructed datasets, and they find that they all generalize very poorly on datasets with maximal compound divergence, and that furthermore the compound divergence appears to be anticorrelated with accuracy.

--

Discussion:

Blind review #1 is the most knowledgeable in this area and wrote "This is an interesting and ambitious paper tackling an important problem. It is worth noting that the claim that it is the compound divergence that controls the difficulty of generalization (rather than something else, like length) is a substantive one, and the authors do provide evidence of this."

--

Recommendation and justification:

This paper deserves to be accepted because it tackles an important problem that is overlooked in current work that is evaluated on datasets of questionable meaningfulness. It adds insight by focusing on the qualities of datasets that enable testing how well learning algorithms do on compositional generalization, which is crucial to intelligence.